# META-RL INDUCES EXPLORATION IN LANGUAGE AGENTS

**Yulun Jiang**[1,2,*]  **Liangze Jiang**[1,3,*]  **Damien Teney**[3]  **Michael Moor**[2,†]  **Maria Brbić**[1,†]

[1]EPFL  [2]ETH Zurich  [3]Idiap Research Institute
[*]*Equal Contribution*  [†]*Equal Advising*

## ABSTRACT

Reinforcement learning (RL) has enabled the training of large language model (LLM) agents to interact with the environment and to solve multi-turn long-horizon tasks. However, the RL-trained agents often struggle in tasks that require active exploration and fail to efficiently adapt from trial-and-error experiences. In this paper, we present LAMER, a general Meta-RL framework that enables LLM agents to actively explore and learn from the environment feedback at test time. LAMER consists of two key components: *(i)* a cross-episode training framework to encourage exploration and long-term rewards optimization; and *(ii)* in-context policy adaptation via reflection, allowing the agent to adapt their policy from task feedback signal without gradient update. Experiments across diverse environments show that LAMER significantly improves performance over RL baselines, with $11\%$, $14\%$, and $19\%$ performance gains on Sokoban, MineSweeper and Webshop, respectively. Moreover, LAMER also demonstrates better generalization to more challenging or previously unseen tasks compared to the RL-trained agents. Overall, our results demonstrate that Meta-RL provides a principled approach to induce exploration in language agents, enabling more robust adaptation to novel environments through learned exploration strategies.

## 1 INTRODUCTION

Recent advances in large language models (LLMs) have shifted from building conversational systems to decision-making agents capable of reasoning about and interacting with their environments (Yao et al., 2023b; Shinn et al., 2023; Wang et al., 2025; Feng et al., 2025). To accomplish the goal, language agents operate in multi-turn, textual observation-action loops, and must adapt quickly using the memory across turns. Central to such adaptation is *exploration*, which allows agents to test uncertain actions, acquire new knowledge, and avoid premature convergence on suboptimal strategies. However, unlike humans that can explore systematically and make fast adaptation in new environments (Wilson et al., 2014), LLM agents do not robustly engage in exploration without substantial interventions (Krishnamurthy et al., 2024).

Recent works have begun to address this limitation by guiding LLMs toward exploratory behaviors at test time. For example, Tajwar et al. (2025) train models offline to distill exploration strategies from trajectories across diverse environments, while Gandhi et al. (2024) induce such strategies from offline search traces. Setlur et al. (2025) train models to learn to explore in-context as a better way of spending test-time compute (Snell et al., 2025). However, these works either focus on single-turn non-agentic reasoning problems, or rely on offline data which limits them to imitation rather than active exploration.

In this work, we take a step toward agents that can *actively explore* their environment, gather feedback, and leverage this experience for more effective exploitation. Since multi-turn tasks often have a sparse success signal after an episode, we consider a multi-episode regime (Shinn et al., 2023) where an episode is the unit of exploration and exploitation. Balancing exploration and exploitation can then be naturally formulated as a cross-episode reinforcement learning (RL) framework. Training across many similar but different environments under this framework leads to the meta reinforcement learning (Meta-RL) (Duan et al., 2016; Wang et al., 2016; Bauer et al., 2023; Beck et al.,

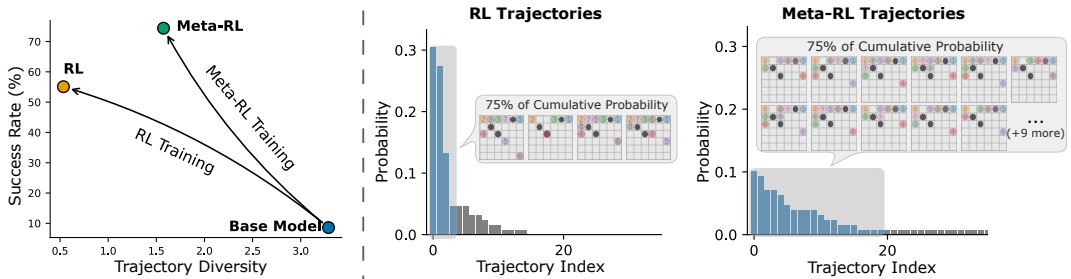

Figure 1: Comparison of RL and Meta-RL training on the MineSweeper environment. *Left*: Meta-RL training with LAMER retains higher sample diversity from the base model while achieving better success rates, reaching a better trade-off between exploration and exploitation. *Right*: Distinct trajectories and their empirical probabilities aggregated over multiple sampled trajectories in the MineSweeper environment. Each trajectory corresponds to a sequence of clicks (numbered cell) on the board. Sample diversity is quantified by the entropy of the empirical distribution. The Meta-RL trained model produces more diverse and explorative trajectories.

2025), where the agent is forced to discover general strategies that work in unseen and potentially harder environments.

Building upon this, we propose LAMER (**L**LM **A**gent with **Me**ta-**R**L), a general Meta-RL framework for LLM agent training. LAMER contains two key design principles. First, unlike standard single-episode RL, LAMER is designed around a *multi-episode structure* to train the agent to solve the problem through trial and error. In early episodes, the agent is encouraged to gather diverse experiences and informative feedback from the environment, which are then used to adapt its policy in later episodes. By maximizing long-term rewards across episodes, the agent internalizes a learning algorithm that explicitly incentivizes exploration for improved downstream exploitation. Second, at both training and test time, the agent effectively leverages the feedback and reflections from previous episodes to determine its strategy for the next episode. This essentially implements an *RL algorithm in context*, making the approach naturally suited for LLM agents. Meta-RL produces more diverse samples while simultaneously achieving higher performance, reaching a better balance between exploration and exploitation than standard RL (Figure 1). To the best of our knowledge, this is the first time a meta-RL framework is used for LLM agent training.

We evaluate LAMER on four challenging long-horizon tasks: Sokoban (Racanière et al., 2017), MineSweeper (Li et al., 2024), Webshop (Yao et al., 2022) and ALFWorld (Shridhar et al., 2021). Using Qwen3-4B (Yang et al., 2025), we demonstrate that LAMER consistently outperforms prompting and RL baselines across all environments. In addition, we observe that the trained model learns to balance between exploration and exploitation, resulting in improved test-time scaling performance. In particular, LAMER adapts the trained policy at test time, with $11/14/19\%$ absolute performance gains on Sokoban, MineSweeper and Webshop, respectively, over the RL. Furthermore, we show that the LAMER trained model generalizes better to harder and out-of-distribution tasks. In summary, LAMER takes a step toward autonomous agents that can actively act to uncover information and improve their decision-making in the new environments.

## 2 RELATED WORK

**LLM-as-agent.** As LLMs become increasingly capable of reasoning about complex scenarios (Wei et al., 2022), there is a growing interest in making them decision-making autonomous agents. Earlier works rely on prompting frozen LLMs (Yao et al., 2023b; Shinn et al., 2023; Park et al., 2023; Wang et al., 2024a; AutoGPT). ReAct (Yao et al., 2023b) prompts LLMs with in-context examples to generate both textual actions and reasoning thoughts. Later, Reflexion (Shinn et al., 2023) extends this principle to the multi-episode setting, where the agent verbally reflects on the last episode and maintains their own reflection buffer for the next episodes. More recent research trains LLM agents through designing advanced RL algorithms (Wang et al., 2025; Feng et al., 2025) for multi-turn interactions, or supervised fine-tuning on generated interaction trajectories across diverse tasks (Tajwar et al., 2025). The evaluation of LLM agents also poses challenges because of fully verbal interactions with the environments. Recent benchmarks span a wide range of domains, including text-based

embodied environments (Shridhar et al., 2021), e-commerce website (Yao et al., 2022), bandits (Nie et al., 2024), classic games (Park et al., 2025; Li et al., 2024) and other tasks (Liu et al., 2024; Nathani et al., 2025). For a more comprehensive overview of these efforts, we refer readers to recent surveys (Wang et al., 2024b; Zhang et al., 2025).

**Meta reinforcement learning** (Meta-RL) (Beck et al., 2025) focuses on "learning to reinforcement learn" in order to rapidly adapt to new environments. Similar to meta-learning (Thrun & Pratt, 1998; Hospedales et al., 2021), it involves an inner-loop that represents an RL algorithm (*i.e.*, an adaptation strategy) by itself, together with an outer-loop that updates the meta-parameters so that the inner loop becomes more effective across many tasks. By training across many tasks, the outer loop forces the agent to learn the exploration strategy necessary to solve the tasks, while the inner loop enables the agent adapt quickly based on the exploration. Depending on how the inner loop is done, there are in-context methods and gradient-based methods. For example, Duan et al. (2016); Wang et al. (2016); Stadie et al. (2018) represent the inner-loop as a history-dependent policy parametrized by an RNN, the adaptation is thus done 'in-context' through gathering more information stored in the memory states. On the other hand, Finn et al. (2017) leverages a gradient-based approach in which the inner adapts a general meta-policy learned by the outer-loop. Our work lies in the former category, where the adaptation occurs entirely in-context at test time, naturally leveraging LLMs' in-context learning abilities.

**Test-time compute.** A Meta-RL framework in LAMER can be considered as amortizing the test-time compute by training tasks in a multi-episode manner rather than a single episode. In this way, the learned in-context policy adaptation balances exploration and exploitation for a fast adaptation at test time. This is essentially a better way of spending test-time compute (Snell et al., 2025; Muennighoff et al., 2025; Wu et al., 2025; Setlur et al., 2025). In our experiments, we match the training compute budget between an RL and a Meta-RL baseline, and show that meta-RL encourages superior test-time scaling behavior (through pass@k). Qu et al. (2025) similarly relates meta-RL to test-time compute, but they are limited to single-turn problems of mathematical reasoning, without leveraging the interactive feedback from the environment.

**Reasoning in LLMs.** More broadly, this work relates to reasoning in LLMs, because language agents must use reasoning as part of their decision-making. A large bulk of recent work on LLM reasoning has focused on more advanced prompting (Wei et al., 2022; Yao et al., 2023a), post-training (Cobbe et al., 2021; Luong et al., 2024; Shao et al., 2024; DeepSeek-AI et al., 2025) or bootstrapping (Zelikman et al., 2022) against verifiers or reward models, inducing structured search behavior (Gandhi et al., 2024; Moon et al., 2024), or reflecting on previous answers (Kumar et al., 2024; Xiong et al., 2025; Qu et al., 2024), etc. Most of these works focus on single-turn math (Hendrycks et al., 2021b; Cobbe et al., 2021) and coding (Chen et al., 2021; Hendrycks et al., 2021a) problems, while we target multi-turn agentic environments where environment feedback is available after every action and at the end of the episode.

## 3 Preliminaries

We consider the scenario where an LLM agent interacts with the environment to solve a multi-turn task. This process can be formulated as a Markov decision process $\mathcal{M} = (\mathcal{S}, \mathcal{A}, P, R, \gamma_{\text{step}})$, where $\mathcal{S}$ and $\mathcal{A}$ denote the state space and action space, and $R$ is the reward function. At each time step $t = 0, ..., T-1$, the LLM agent observes a state $s_t \in \mathcal{S}$ and selects an action $a_t \in \mathcal{A}$ according to its policy $a_t \sim \pi_\theta(\cdot|s_t)$. The environment then provides a scalar reward $r_t \in \mathbb{R}$ and transitions to the next state $s_{t+1}$ according to the transition function $P(\cdot \mid s_t, a_t)$. A trajectory is the sequence of states, actions, and rewards over an episode, *i.e.*, $\tau = (s_0, a_0, r_0, ..., s_{T-1}, a_{T-1}, r_{T-1})$. The objective of reinforcement learning is to maximize the expected discounted return:

$$\mathbb{E}_{\tau \sim \pi_\theta} \left[ \sum_{t=0}^{T-1} \gamma_{\text{step}}^t r_t \right], \tag{1}$$

where $\gamma_{\text{step}} \in [0, 1]$ is the discount factor. Recent works (Wang et al., 2025; Feng et al., 2025) have shown that RL training has enabled LLM agents to interact with the environment and solve multi-turn tasks. However, such agents often learn a fixed policy during training and struggle to actively explore and adapt their behavior to the tasks at *test time* (Nie et al., 2024).

**Meta-RL.** Conversely, by training on a distribution of tasks, Meta-RL (Duan et al., 2016; Wang et al., 2016; Bauer et al., 2023; Beck et al., 2025) encourages exploration because it optimizes meta-parameters, such that the agent can solve new tasks quickly. In our case, the meta-parameters are the parameters of the LLM. This necessitates the agent to learn general exploration-exploitation strategies suitable for the task distribution trained on. For example, for most navigation tasks in partially observable environments, the optimal strategy is to gather the environment information and locate the target during the first episode, then reach the target as efficiently as possible in the second episode. This *explore-then-exploit* strategy implemented by the agent is itself an RL algorithm, where the policy learned at the meta-level encodes how to adaptively switch between information-gathering and reward-maximizing behaviors depending on the stage of interaction with a new task. For LLM agents operating in multi-turn tasks, the policy can be in context (*i.e.*, without parameter updates at test time), naturally leveraging the in-context capability of LLMs.

## 4 LaMer: A Meta-RL Framework for LLM agents

Adopting the Meta-RL principle, we present LaMer, a framework for training LLM agents with the ability to actively explore and adaptively learn from the environment. The framework addresses two central challenges: *(i)* how to balance exploration and exploitation over multiple attempts at a task, and *(ii)* how to efficiently adapt the policy during training and evaluation. To address the first challenge, LaMer introduces a *cross-episode training scheme* that treats each trial as a sequence of episodes, enabling the agent to explore in early episodes and exploit this information in later ones. Second, instead of relying on gradient-based updates, LaMer uses *self-reflection as an in-context adaptation mechanism*, allowing the agent to summarize past experiences and adjust its strategy accordingly. Together, these two components enable scalable training of LLM agents under a unified Meta-RL framework, which can be optimized with standard RL algorithms.

**Cross-episode training framework.** In the training of LaMer, each trial consists of $N$ episodes sequentially generated by the agent:

$$\mathcal{T} = (\tau^{(0)}, \tau^{(1)}, \ldots, \tau^{(N-1)}), \quad \text{where } \tau^{(n)} \sim \pi_\theta^{(n)}(\cdot), \ n \in [0, N-1], \tag{2}$$

where $\pi_\theta^{(n)}(\cdot)$ is the policy at episode $n$ updated from the accumulated history $\tau^{(0)}, \ldots, \tau^{(n-1)}$ through some adaptation strategy. For simplicity, in our analysis we assume all episodes contain the $T$ steps of interactions with the environment, *i.e.*, $\tau^{(n)} = (s_0^{(n)}, a_0^{(n)}, r_0^{(n)}, \ldots, s_{T-1}^{(n)}, a_{T-1}^{(n)}, r_{T-1}^{(n)})$ for all $n \in [0, N-1]$. The rollout process terminates at $n$ if $\tau^{(n)}$ is successful (as indicated by the environment feedback). Otherwise, the agent starts a new episode $\tau^{(n+1)}$ from the same initial state, repeating this procedure until the maximum episode budget is reached. For action $a_t^{(n)}$, the discounted return $g_t^{(n)}$ *within the episode* $\tau^{(n)} \in \mathcal{T}$ is:

$$g_t^{(n)} = \sum_{l=t}^{T-1} \gamma_{\text{step}}^{l-t} r_l^{(n)}, \tag{3}$$

where $\gamma_{\text{step}} \in [0, 1]$ is within-the-episode discount factor.

To enhance the exploration and maximize the long-term reward, in LaMer framework we define the discounted return $G_t^{(n)}$ *across the episodes* of $\mathcal{T}$ as:

$$G_t^{(n)} = \underbrace{g_t^{(n)}}_{\text{within-the-episode}} + \underbrace{\sum_{m=n+1}^{N-1} \gamma_{\text{traj}}^{m-n} g_0^{(m)}}_{\text{cross-episode}}, \tag{4}$$

where $\gamma_{\text{traj}} \in [0, 1]$ is the cross-episode discount factor. Finally, the LLM agent is trained via the following Meta-RL objective:

$$J(\theta) = \mathbb{E}_{\mathcal{T} \sim \pi_\theta} \left[ \sum_{n=0}^{N-1} \gamma_{\text{traj}}^n \sum_{t=0}^{T-1} \gamma_{\text{step}}^t r_t^{(n)} \right] = \mathbb{E}_{\mathcal{T} \sim \pi_\theta} \left[ G_0^{(0)} \right]. \tag{5}$$

Here, $\gamma_{\text{traj}}$ is an important factor for the trade-off between *exploration* and *exploitation*. Ideally, small $\gamma_{\text{traj}}$ biases the objective towards early episodes and will lead to rapid exploitation to solve the problem. In comparison, a larger $\gamma_{\text{traj}}$ emphasizes long-horizon return and therefore encourages more exploration at the early stage.

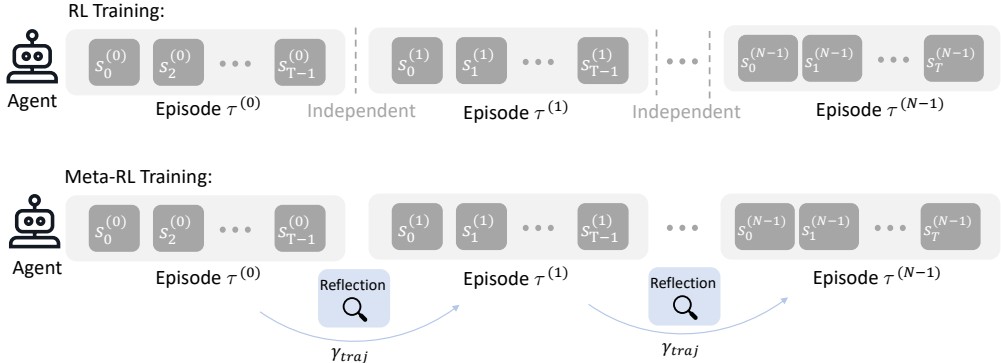

Figure 2: Comparison between the training processes of RL (top) and Meta-RL used in LAMER (bottom). For a single task, RL generates a group of trajectories independently. In contrast, in LAMER we use Meta-RL and produce the trajectories sequentially and adapt the policy in-context with self-reflection. Trajectory discount factor $\gamma_{\text{traj}}$ is used for cross-episode credit assignment.

**In-context policy adaptation with self-reflection.** In the Meta-RL, policy adaptation is the inner loop of the learning process of an LLM agent. Therefore, a flexible and efficient adaptation mechanism plays an important role during training and methods like gradient descent (Finn et al., 2017) might be too expensive, especially for LLMs. In LAMER, we propose a self-reflection based strategy(Shinn et al., 2023) to *adapt the policy in-context* (Brown et al., 2020; Laskin et al., 2023). Specifically, after each episode finishes, we prompt the agent to generate the textual reflection on the previous attempt, providing specific feedback and plan to guide the next episode (see Appendix A for the used prompt). The policy is therefore updated through modifying the context, $\pi_\theta^{(n)}(\cdot) = \pi_\theta(\cdot|\mathcal{H}^{(n)})$ where $\mathcal{H}^{(n)}$ denotes the inter-episode memory that contains both the history trajectories and reflections. Importantly, the self-reflection step is also explicitly trained in LAMER using the reward obtained in the next episode. Note that the content $\mathcal{H}^{(n)}$ can be adjusted according to the predefined memory buffer to reduce the context length and improve the efficiency. By default, we retain both history and reflection in $\mathcal{H}^{(n)}$, and provide an ablation study in Section 6.2.

**Comparison to the RL training.** Compared to the RL objective (Eq. 1), Meta-RL extends the credit assignment across multiple episodes to incentivize exploration in the early stages. In practice, given a single task, both RL and Meta-RL will sample a group of episodes during training to estimate the advantage. The key difference is that the RL rollouts are independent, whereas in Meta-RL each episode is conditioned on the preceding rollouts within the trial. Figure 2 illustrates the conceptual difference between the training processes of RL and Meta-RL.

**Optimization.** The proposed Meta-RL objective in Eq. (5) can be optimized with standard policy gradient methods. Given the per-action cross-episode return $G_t^{(n)}$ defined above, the gradient can be estimated by

$$\nabla_\theta J(\theta) = \mathbb{E}_{\mathcal{T} \sim \pi_\theta} \left[ \sum_{n=0}^{N-1} \sum_{t=0}^{T-1} \nabla_\theta \log \pi_\theta(a_t^{(n)}|s_t^{(n)}, \mathcal{H}^{(n)}) A_t^{(n)} \right], \quad (6)$$

where $A_t^{(n)}$ is the advantage estimation derived from $G_t^{(n)}$. The framework is compatible with widely used optimizers such PPO (Schulman et al., 2017) and critic-free approaches such as GRPO (Shao et al., 2024) and GiGPO (Feng et al., 2025).

## 5 EXPERIMENTS

In this section, we conduct comprehensive experiments to evaluate LAMER in the Meta-RL framework. Specifically, we present the evaluation on: *(i)* the overall performance of LAMER across different agent environments; *(ii)* the generalization ability of LAMER to harder tasks; and *(iii)* the generalization of LAMER under distribution shifts.

### 5.1 EXPERIMENTAL SETUP

**Environments.** We evaluate LAMER on four challenging and diverse environments: Sokoban (Racanière et al., 2017), MineSweeper (Li et al., 2024), Webshop (Yao et al., 2022) and ALFWorld (Shridhar et al., 2021). Among them, Sokoban is a classic grid-based game on planning where the environment is *fully observable*. In comparison, the environments of MineSweeper, ALFWorld and Webshop are *partially observable*, requiring the agent to explore and plan under uncertainty to finish the task. Specifically, MineSweeper is a board game about logical deduction on hidden cells. Webshop simulates realistic web-based shopping tasks, and ALFWorld provides text-based embodied environments. We provide the detailed explanation and prompts of the environment in Appendix A. All the experiments are conducted with the text modality, though the proposed method can be naturally applied to multimodal environments.

**Training details.** We use Qwen3-4B (Yang et al., 2025) as our base model for all the experiments. To improve rollout efficiency in agentic loops, we use the non-thinking mode during trajectory generation. Additionally, we validate our method on Llama3.1-8B-Instruct (Grattafiori et al., 2024), see Appendix D.1 for the results. For the Meta-RL setting, we use $\gamma_{\text{traj}} = 0.6$ as the default trajectory discount factor and explore its influence in the ablation study. We use GiGPO as the default optimization algorithm for all the environments with LAMER. Importantly, for Meta-RL training we sample $N = 3$ episodes and set group size to 8 for each task. To ensure a fair comparison, we use a group size of 24 in standard RL training, yielding the same number of trajectories used for each gradient update step. All other hyperparameters and configurations are kept identical across RL and Meta-RL for a fair comparison and are provided in Appendix C. Our code is publicly available at `https://github.com/mlbio-epfl/LaMer`.

### 5.2 PERFORMANCE COMPARISON

We compare the performance of the LAMER framework with prompting-based baselines (Zero-shot, ReAct (Yao et al., 2023b; Shinn et al., 2023)), and RL methods (PPO (Schulman et al., 2017), RLOO (Ahmadian et al., 2024), GRPO (Shao et al., 2024), and GiGPO (Feng et al., 2025)) across three environments: Sokoban, MineSweeper, and Webshop. For each method, we report the success rates under 1, 2, and 3 attempts (*i.e.*, pass@1, pass@2, and pass@3, respectively). The results are summarized in Table 1.

Table 1: Performance on Sokoban, MineSweeper and Webshop environments. The results of p@1, p@2 and p@3 denote the success rate (%) under 1, 2, and 3 attempts, respectively.

| Method | Sokoban | | | MineSweeper | | | Webshop | | |
|---|---|---|---|---|---|---|---|---|---|
| | p@1 | p@2 | p@3 | p@1 | p@2 | p@3 | p@1 | p@2 | p@3 |
| *Prompting* | | | | | | | | | |
| Zero-shot | 6.8 | 9.8 | 12.9 | 4.5 | 6.6 | 8.6 | 1.4 | 2.1 | 2.3 |
| ReAct | 7.2 | 9.6 | 12.5 | 6.3 | 7.0 | 10.9 | 3.1 | 4.5 | 4.5 |
| Reflexion | 6.4 | 9.8 | 12.1 | 5.5 | 7.2 | 9.8 | 2.7 | 3.3 | 3.5 |
| *Training with RL* | | | | | | | | | |
| PPO | 12.5 | 15.4 | 16.8 | 29.7 | 34.2 | 35.5 | 53.1 | 54.5 | 54.9 |
| RLOO | 13.5 | 16.6 | 18.8 | 48.8 | 51.2 | 51.6 | 67.6 | 68.4 | 69.1 |
| GRPO | 22.9 | 26.4 | 27.0 | 36.3 | 40.0 | 40.4 | 72.9 | 73.0 | 73.0 |
| GiGPO | 41.6 | 43.6 | 44.1 | **52.0** | 54.9 | 55.1 | **73.4** | 74.6 | 75.2 |
| *Training with Meta-RL (ours)* | | | | | | | | | |
| LAMER | **42.4** | **52.0** | **55.9** | 44.1 | **66.4** | **74.4** | 67.8 | **84.4** | **89.1** |

**Meta-RL obtains better performance.** Across all three environments, LAMER trained with Meta-RL consistently outperforms both prompting-based baselines and RL-training methods on the final pass@3 success rate. On the Sokoban environment, LAMER achieves a $55.9\%$ pass@3 success rate, substantially outperforming the $44.1\%$ from the strongest RL baseline (GiGPO) and $12.9\%$ from prompting-based methods. Similarly, on the MineSweeper environment LAMER reaches $74.4\%$ pass@3 success rate, which is $19\%$ higher than the best RL-trained model. On the Webshop environment, LAMER also performs $14\%$ better than the RL-trained methods. Notably, the performance gains are not limited to pass@3: improvements are also observed on pass@2 for all the environments, and even pass@1 for Sokoban. Together, these results demonstrate that LAMER delivers consistent benefit on the trained agents to solve the long-horizon task in the complex environments.

**Meta-RL exhibits stronger test-time scaling.** LAMER trained with Meta-RL demonstrates remarkable effectiveness in test-time scaling, with larger performance gains across attempts according to the results in Table 1. For example, from pass@1 to pass@3 on Sokoban LAMER achieves $13.5\%$ improvement, significantly larger than both RL-trained and prompting-based baselines (which are less than 5%). Notably, although LAMER starts with slightly lower pass@1 performance than the RL baseline (GiGPO) in the MineSweeper and Webshop environments, it quickly recovers and surpasses all baselines by pass@2 and pass@3. The results indicate that the Meta-RL trained model has successfully learned to actively explore in the earlier episodes and adapt effectively from the mistakes, leading to significant gains in the subsequent attempts. The illustrative trajectories and reflections produced by the trained agents are presented in Appendix E.

**Meta-RL induces exploration.** To further analyze the behavior of the models, we measure the diversity of answer trajectories across environments. For each question, we sample multiple trajectories from the agent and group the identical trajectories that have the same states and actions. These groups are used to form the empirical distribution over distinct trajectories, as shown in Figure 1. We then estimate the entropy the distribution to quantify the *trajectory diversity*. Figure 3 compares the trajectory diversity of the base model, RL, and Meta-RL agents across environments. We observe that the base model exhibits the highest entropy, indicating it generates a wide range of trajectories, though this diversity does not translate into higher success rates (see Table 1). RL-trained agent reduces diversity and converges toward more deterministic behaviors. In contrast, LAMER trained with Meta-RL preserves a higher level of diversity than RL baselines, allowing more exploration at test time.

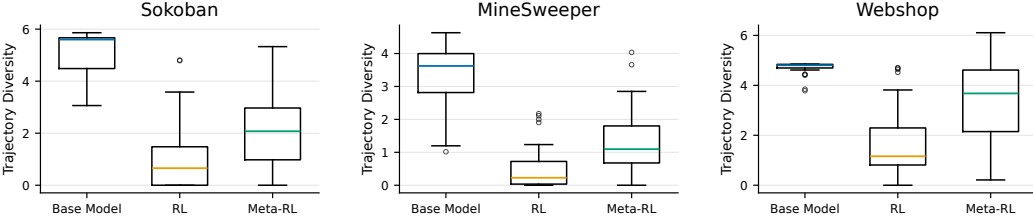

Figure 3: Trajectory diversity of base and trained models. Compared to RL, Meta-RL preserves more diverse trajectories from the base model, striking a better balance between exploration and exploitation.

## 5.3 GENERALIZATION TO HARDER TASKS

Next we study the generalization ability of the pretrained models on harder tasks. To this end, we take the models trained with the RL and Meta-RL and evaluate them on the harder tasks in the environments of Sokoban and MineSweeper. We increase the difficulty by using more boxes for Sokoban and more mines for MineSweeper in the grid. The results are shown in Figure 4. As expected, the model trained with both RL and Meta-RL underperforms on harder tasks with an increasing number of boxes or mines in the grid. However, Meta-RL consistently outperforms RL on all the difficulty levels. Notably, on the most difficult setting, the model trained from Meta-RL still outperforms the RL-trained model with $10\%$ performance gap on Sokoban, and $5\%$ performance

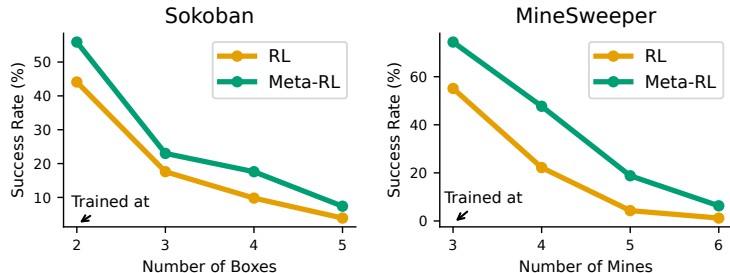

Figure 4: Performance of RL and Meta-RL trained model on the tasks with increased difficulty. For Sokoban, we gradually increase the number of boxes and for MineSweeper, we increase the number of mines in the grid.

gap on the MineSweeper. The consistent gap indicates that LaMer trained with Meta-RL not only performs better on the training distribution, but also generalizes better to the harder tasks.

## 5.4 GENERALIZATION TO UNSEEN TASKS

We further study the ability LaMer and alternative methods to generalize out-of-distribution. For this experiment, we use the ALFWorld environment (Shridhar et al., 2021). As a text-based embodied environment, ALFWorld contains 6 categories of common household activities: Pick and Place (*Pick*), Examine in Light (*Look*), Clean and Place (*Clean*), Heat and Place (*Heat*), Cool and Place (*Cool*), and Pick Two and Place (*Pick2*). We use the tasks of *Pick*, *Look*, *Clean* and *Heat* as in-distribution tasks and use *Cool* and *Pick2* as out-of-distribution tasks. We train LaMer and alternative baselines with instances from in-distribution tasks and then evaluate the model on both in-distribution tasks (with held-out test set), and the examples of out-of-distribution tasks. The results are shown in Table 2. As we can see, RL trained model generally performs well on in-distribution tasks and outperforms prompting-based methods by achieving more than 20% improvement on *Look*, *Clean* and *Heat*. However, on out-of-distribution tasks, *Cool* and *Pick2*, it only obtains 58.1% and 36.0% success rate. LaMer with Meta-RL consistently outperforms RL on both in-distribution and out-of-distribution tasks, with a notable performance gap on out-of-distribution tasks. In particular, our LaMer framework achieves 23% performance gains on Cool and around 14% on *Pick2*. Overall, these results suggest that on ALFWorld, Meta-RL trained model could generalize better to out-of-distribution tasks compared to the RL trained model.

Table 2: Evaluation of out-of-distribution generalization on the tasks of ALFWorld.

| Method | i.d | | | | o.o.d | |
|---|---|---|---|---|---|---|
| | Pick | Look | Clean | Heat | Cool | Pick2 |
| Prompting | 91.9 | 52.9 | 48.4 | 44.8 | 42.8 | 21.2 |
| RL | 95.5 | 83.0 | 67.9 | 86.6 | 58.1 | 36.0 |
| Meta-RL | **97.7** | **100.0** | **90.2** | **89.5** | **81.0** | **50.2** |

## 6 ANALYSIS

We further conduct a series of ablation studies on the key design factors of LaMer, including *(i)* the influence of trajectory discounted factor $\gamma_{\text{traj}}$ on the trade-off between exploration and exploitation and *(ii)* the ablation of inter-episode memory configurations. We additionally discuss *(iii)* the computation budget of the proposed Meta-RL framework compared to the RL training.

### 6.1 INFLUENCE OF TRAJECTORY DISCOUNT FACTOR

The cross-episodes discount factor $\gamma_{\text{traj}}$ controls how rewards are propagated within a trial, thereby mediating the balance between exploration and exploitation in the LaMer framework during training. To understand the effect of the discount factor, we train the agents with LaMer using different values of $\gamma_{\text{traj}}$ on Sokoban, MineSweeper and Webshop environments (Figure 5). We observe that a

larger value of $\gamma_{\text{traj}}$ does not necessarily lead to better final performance on pass@3, instead, the optimal setting of $\gamma_{\text{traj}}$ varies across different environments. For Sokoban and Webshop environments, intermediate values (*e.g.*, $\gamma_{\text{traj}} = 0.6$) yield the best results, suggesting that balancing immediate and long-term rewards is more important for these tasks. In contrast, MineSweeper benefits from relatively larger $\gamma_{\text{traj}}$ (*e.g.*, $\gamma_{\text{traj}} = 0.9$), indicating that extended credit assignment better supports strategic exploration in this environment. Overall, the results show that $\gamma_{\text{traj}}$ provides a practical way to control the trade-off between exploration and exploitation across environments.

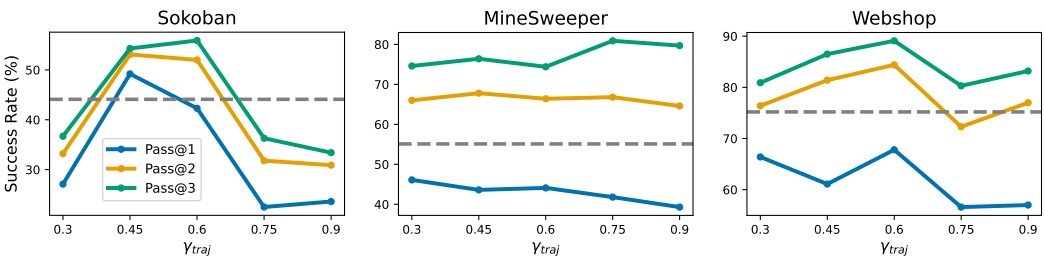

Figure 5: Success rates of models trained with different $\gamma_{\text{traj}}$. A higher value encourages more exploration during training.

## 6.2 ABLATION ON THE INTER-EPISODE MEMORY

In LAMER, the agent policy is adapted *in-context* through the inter-episode memory $\mathcal{H}^{(n)}$, which by default contains both the trajectories and reflections of previous episodes. To assess the influence of memory content to the training, we consider two alternative configurations in $\mathcal{H}^{(n)}$: *(i)* only history trajectories; *(ii)* only reflections. The performance of the trained agents in each configurations are reported in Table 3. The results show that self-reflection provides a clear benefit in LAMER, leading to $21.6\%$ improvement on Sokoban, $11.0\%$ on MineSweeper and $3.5\%$ on Webshop, respectively. Interestingly, the reflection-only configuration also outperforms the default setting in LAMER (which contains both trajectory and reflection) across all environments. We hypothesize that this is because reflection-only memory presents more concise and focused guidance, leading to more effective adaptation of the agent's behavior.

Table 3: Comparison of LAMER with different inter-episode memory configurations.

| Content in $\mathcal{H}^{(n)}$ | Sokoban | MineSweeper | Webshop |
|---|---|---|---|
| Trajectory-only | 34.8 | 69.5 | 89.3 |
| Reflection-only | **56.4** | **80.5** | **92.8** |
| Both | 55.9 | 74.4 | 89.1 |

## 6.3 TRAINING BUDGET

We next analyze the training budget of RL and Meta-RL, focusing on both data usage and computational efficiency. To ensure a fair comparison, we set the group size for standard RL to be three times larger than that of Meta-RL. This adjustment guarantees that the two methods consume the same number of trajectories for each gradient update. Aside from this scaling, all other experimental configurations—such as learning rates, batch sizes, and network architectures—are held constant. This design choice highlights that Meta-RL does not require a larger data budget compared to RL; in other words, both methods rely on the same total number of trajectories to learn.

Nevertheless, LAMER might still introduce additional training time cost compared to the RL baselines. In RL training, all the episodes could be sampled in parallel since they are independent. In contrast, LAMER exhibits less parallelism since episodes within the same trial needs to be generated sequentially. As a result, we observe around twice the training time cost for LAMER in our current implementation. This suggests that more sophisticated sampling strategies, such as asynchronous rollout, could further improve the efficiency of LAMER for training LLM agents.

# 7 CONCLUSION

Being able to explore and gather information from the environment is crucial in building autonomous agents that can adapt quickly and robustly. We introduced LAMER, a general LLM agent training framework leveraging the principle of meta reinforcement learning. Unlike previous RL methods that maximize a single-episode return for immediate payoff, LAMER maximizes a discounted cross-episode return, naturally balancing when to explore versus when to exploit to maximize long-term performance. The exploration allowed at training time teaches the agent general explorative strategies that enable a rapid in-context adaptation at test time. We show across diverse environments that LAMER significantly outperforms RL methods, is able to generalize to harder environments, and scales better with more episodes at test time.

**Limitations and future work.** Our results raise several promising directions for future work. *(i)* The generality of our method allows for combining it with other RL algorithms or self-reflection frameworks. We hypothesize that a more advanced advantage estimation strategy or a stronger reasoning model may enhance the performance. *(ii)* Our approach requires sampling episodes sequentially for rollouts since episodes are dependent in cross-episode training. This eventually leads to longer training time than RL methods. More efficient training strategies will be explored in future work. *(iii)* Finally, LAMER trained on easier environments can generalize to harder environments of the same kind or relatively similar domains. This ultimately suggests possibilities in building generalist agents that can adapt to completely novel environments.

## ACKNOWLEDGEMENTS

M.B. gratefully acknowledges the support of the Swiss National Science Foundation (SNSF) starting grant TMSGI2_226252/1, SNSF grant IC00I0_231922, and the Swiss AI Initiative. M.B. is a CIFAR Fellow in the Multiscale Human Program.

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

APPENDIX

## A    Task Description and Details

**Sokoban.** We include the classic video game Sokoban as a fully-observable environment. The game is a 2D square board, with $N$ boxes scattered on the board. There are also $N$ target positions marked on the board. The player is placed at an initial position, and the goal is to push all the boxes to the target positions. There is no correspondence between each box and the target position. When the player walks into a box, it gets pushed in that direction (if there's space). Boxes can't be pushed into walls or other boxes. Once a box is pushed into a corner or against a wall with no way to get behind it, it might become permanently stuck. There is no pull operation in this game. The agent, therefore, has to think several moves ahead to avoid getting boxes stuck in positions where they can't reach their targets. The difficulty of this task is controlled by the board size, the number of boxes, and the wall structure of the board. We train on a board size $6 \times 6$ with 2 boxes.

**MineSweeper.** We include the classic video game MineSweeper as a partially-observable environment. The game is a 2D square board, with several mines randomly scattered in the board cells. The goal of the game is to open all the safe cells without revealing the hidden mines. In each step, the agent opens a cell, and the first step is always safe. If a mine is revealed, the task ends in failure immediately. The state of the opened (safe) cells can either be empty or a number from 1 to 8, and the number specifies how many mines are adjacent to the specific cell. The agent needs to use the numbers marked on the opened cells to reason about the position of mines. Success is achieved when all safe cells are revealed. Our implementation is based on a simplified version of Li et al. (2024). The difficulty of this task is controlled by the board size and the number of mines. We train on a board size $6 \times 6$ with 3 hidden mines.

**Webshop** (Yao et al., 2022). We include Webshop as a partially-observable text-based environment that simulates online shopping. The agent is given a natural language instruction specifying a product to purchase with certain attributes. The environment presents a simplified e-commerce interface where the agent can search for products, navigate through search results, and examine product pages with details like price, color, size, and customer reviews. The agent must interpret the instruction, search effectively, filter through multiple product options, and select the item that best matches the specified criteria. Success is measured by whether the final purchased item satisfies all the requirements in the original instruction.

**ALFWorld** (Shridhar et al., 2021). We include ALFWorld as a partially-observable text-based environment that simulates household tasks in interactive fiction format. The agent receives natural language instructions for common household activities. The environment provides text descriptions of rooms, objects, and possible actions, while the agent must navigate through a house, interact with objects, and complete multi-step tasks. Objects may need to be found, picked up, cleaned, heated, or combined with other objects to achieve the goal. The agent's view is limited to the current room and nearby objects, requiring exploration and memory of previously visited locations. Success requires understanding the instruction, planning a sequence of actions, and executing them correctly while managing partial observability. We train on the training examples of the activities 'Pick', 'Look', 'Clean', 'Heat'. We evaluate in-distribution on the test examples from the same activities, and we evaluate out-of-distribution on the test examples from 'Cool', 'Pick2' activities.

## B  EXAMPLE PROMPTS

We provide examples of prompts for each task. There are two types of prompts: (1) the standard version (with name 'Standard Prompt') used for prompting the agent to play the game; (2) the reflection prompt used for self-reflection on a past experience (with name 'Reflection Prompt')

There are variables such as {past_experience_reflection}, {history_actions} in the prompts, among with other task-specific hyperparameters. They are omitted in the prompts for clarity. In practice, they will be replaced with the actual content. Note that the {past_experience_reflection} will be empty for the first episode.

Similar to (Feng et al., 2025), we use <action> </action> block to indicate the final decision of the action, and we use <remark> </remark> to indicate the content of the self-reflection.

(see next page for the prompts)

## B.1 SOKOBAN

---

**Sokoban Standard Prompt**

You are an expert agent operating in the Sokoban environment.

# Symbols and Their Meaning
- Walls (#): These block movement. You can't move through or push anything into walls.
- Floor (_): Open spaces where you can walk and move boxes.
- Targets (O): The spots where boxes need to go.
- Boxes (X): These are what you need to push onto the targets.
- Player (P): That's you! You'll move around the grid to push boxes.
- Box on Target (✓): A box successfully placed on a target.
- Player on Target (S): You standing on a target.

# Goal
Your goal is to push all the boxes (X) onto the target spots (O). Once all boxes are on the targets, you win!

# Rules
Your admissible actions are ["up", "down", "left", "right"].
You can only push one box at a time. You can't pull boxes, so plan ahead to avoid getting stuck.
You can't walk through or push boxes into walls (#) or other boxes.
To avoid traps, do not push boxes into corners or against walls where they can't be moved again.
{example}

# Observations
The initial state of the game is:

```
0:  #  #  #  #  #  #
1:  #  #  #  _  O  #
2:  #  _  O  _  _  #
3:  #  _  _  X  X  #
4:  #  _  #  P  _  #
5:  #  #  #  #  #  #
```

{past_experience_reflection}
You have already taken the following actions:
{history_actions}
Your current observation is:

```
0:  #  #  #  #  #  #
1:  #  #  #  _  O  #
2:  #  _  O  _  X  #
3:  #  _  X  P  _  #
4:  #  _  #  _  _  #
5:  #  #  #  #  #  #
```

Now it's your turn to make moves (choose the next {num_actions_per_turn} actions).
- Your response first be step-by-step reasoning about the current situation — observe the positions of boxes and targets, plan a path to push a box toward a target, and avoid traps like corners or walls.
- Then choose {num_actions_per_turn} admissible actions and present them within `<action> </action>` tags (separated by comma).

---

---

**Sokoban Reflection Prompt**

You are an expert agent operating in the Sokoban environment.

# Symbols and Their Meaning
- Walls (#): These block movement. You can't move through or push anything into walls.
- Floor (_): Open spaces where you can walk and move boxes.
- Targets (O): The spots where boxes need to go.
- Boxes (X): These are what you need to push onto the targets.
- Player (P): That's you! You'll move around the grid to push boxes.
- Box on Target (✓): A box successfully placed on a target.
- Player on Target (S): You standing on a target.

# Your Goal
Your goal is to push all the boxes (X) onto the target spots (O). Once all boxes are on the targets, you win!

# Rules
Your admissible actions are ["up", "down", "left", "right"].
You can only push one box at a time. You can't pull boxes, so plan ahead to avoid getting stuck.
You can't walk through or push boxes into walls (#) or other boxes.
To avoid traps, do not push boxes into corners or against walls where they can't be moved again.

# Your Task
You will be given the history of a past experience.
Your job is to **reflect on the past sequence**, identify any **mistakes or inefficiencies**, and then devise a **concise, improved plan** starting from the original initial state.

# Past Experience
The initial state of the game is:

```
0:  #  #  #  #  #  #
1:  #  #  #  _  O  #
2:  #  _  O  _  _  #
3:  #  _  _  X  X  #
4:  #  _  #  P  _  #
5:  #  #  #  #  #  #
```

You have taken the following actions:
{history_actions}
The final state is:

```
0:  #  #  #  #  #  #
1:  #  #  #  _  O  #
2:  #  _  O  _  X  #
3:  #  _  X  P  _  #
4:  #  _  #  _  _  #
5:  #  #  #  #  #  #
```

The task is NOT successfully completed.
Now it's your turn to reflect on the past experience and come up with a new plan of action.
- Your response should first be step-by-step reasoning about the strategy and path you took to attempt to complete the task. Identify where things went wrong or could be better.
- Then devise a concise, new plan of action that accounts for your mistake with reference to specific actions that you should have taken.
- Finally, end the response with your reflection and improved plan inside `<remark>` `</remark>` tags, to guide the next trial.

## B.2 MINESWEEPER

---

**MineSweeper Standard Prompt**

You are an expert agent operating in the Minesweeper game.
You will be given a two dimensional {board_size} by {board_size} board, with {n_mines} hidden mines.
The rows and columns are indexed from 1 to {board_size}.

# Cell States
- Unopened cells (?): cells that are yet to be revealed and may contain a mine.
- Blank cells (.): opened and non-mine cells, and they have no neighboring mines
- Numbered cells (1-8): opened and non-mine cells, and the number indicates how many mines are in the eight neighboring cells, including those diagonally adjacent. For example, a cell with a '8' means all its neighboring cells contain mines.
- Mine cells (*): opened cells that contain a mine.

# Your Goal
Your goal is to clear the board by revealing all the cells that don't contain mines, without detonating any of the hidden mines scattered throughout the board.
Use clues about the number of neighboring mines in each field to reason about the position of mines and non-mine cells.

# Reveal Rules
Your admissible action is to choose ONE unopened cell (?) to reveal per turn. The outcome depends on the content of that cell:
- Blank cell (.): That cell is revealed, and all contiguous blank cells plus their bordering numbered cells are automatically revealed (auto-cascade).
- Numbered cell (1–8): Only that single cell is revealed, showing the count of neighboring mines.
- Mine (*): The game ends immediately in a loss.
# Observation
The initial state of the game is:

```
Row 1: . . . . . .
Row 2: . . . 1 1 1
Row 3: . . . 1 ? ?
Row 4: 1 1 . 1 2 ?
Row 5: ? 1 . . 1 1
Row 6: ? 1 . . . .
```

{past_experience_reflection}
You have already chosen the following cells to reveal: (6, 1)
Your current observation is:

```
Row 1: . . . . . .
Row 2: . . . 1 1 1
Row 3: . . . 1 ? ?
Row 4: 1 1 . 1 2 ?
Row 5: ? 1 . . 1 1
Row 6: 1 1 . . . .
```

Now it's your turn to make a move.
- Your should first reason step-by-step about the current situation — observe the status of the board, inferring the states of unopened cells (?).
- Then choose ONE unopened cell (?) to reveal. Put the index of cell in the format of "(row, col)" within the `<action> </action>` tag.

---

---

### MineSweeper Reflection Prompt

You are an expert agent operating in the Minesweeper game.
You will be given a two dimensional {board_size} by {board_size} board, with {n_mines} hidden mines.
The rows and columns are indexed from 1 to {board_size}

# Cell States
- Unopened cells (?): cells that are yet to be revealed and may contain a mine.
- Blank cells (.): opened and non-mine cells, and they have no neighboring mines
- Numbered cells (1-8): opened and non-mine cells, and the number indicates how many mines are in the eight neighboring cells, including those diagonally adjacent. For example, a cell with a '8' means all its neighboring cells contain mines.
- Mine cells (*): opened cells that contain a mine.

# Your Goal
Your goal is to clear the board by revealing all the cells that don't contain mines, without detonating any of the hidden mines scattered throughout the board.
Use clues about the number of neighboring mines in each field to reason about the position of mines and non-mine cells.

# Reveal Rules
Your admissible action is to choose ONE unopened cell (?) to reveal per turn. The outcome depends on the content of that cell:
- Blank cell (.): That cell is revealed, and all contiguous blank cells plus their bordering numbered cells are automatically revealed (auto-cascade).
- Numbered cell (1–8): Only that single cell is revealed, showing the count of neighboring mines.
- Mine (*): The game ends immediately in a loss.

# Your Task
You will be given the history of a past experience.
Your job now is to **reflect on the past experience**, identify any **mistakes or inefficiencies**, and then devise a **concise, improved plan** for your next try starting from the original initial state.
# Past Experience
The initial state of the game is:

```
Row 1: . . . . . .
Row 2: . . . 1 1 1
Row 3: . . . 1 ? ?
Row 4: 1 1 . 1 2 ?
Row 5: ? 1 . . 1 1
Row 6: ? 1 . . . .
```

You have chosen the following cells to reveal:
{history_actions}
The final state is:

```
Row 1: . . . . . .
Row 2: . . . 1 1 1
Row 3: . . . 1 ? ?
Row 4: 1 1 . 1 2 ?
Row 5: * 1 . . 1 1
Row 6: 1 1 . . . .
```

The task is NOT successfully completed.
Now it's your turn to reflect on the past experience and come up with a new plan of action.
- Your response should first be step-by-step reasoning about the strategy and path you took to attempt to complete the task. Identify where things went wrong or could be better.
- Then devise a concise, new plan of action that accounts for your mistake with reference to specific actions that you should have taken.
- Finally, end the response with your reflection and improved plan inside `<remark></remark>` tags, to guide the next trial.

## B.3 WEBSHOP

---

**Webshop Standard Prompt**

You are an expert autonomous agent operating in the Webshop e-commerce environment.
Your task is to: Find me slip resistant, non slip men's loafers & slip-ons with rubber outsole, rubber sole with color: 1877blue, and size: 11.5, and price lower than 70.00 dollars.
{past_experience_reflection}{history_actions}
Your admissible actions of the current situation are:
'search[your query]',
'click[search]'.
Now it's your turn to take one action for the current step.
Your response should first be step-by-step reasoning about the current situation, then think carefully which admissible action best advances the shopping goal.
Once you've finished your reasoning, you should choose an admissible action for current step and present it within `<action>` `</action>` tags.

---

**Webshop Reflection Prompt**

You are an expert autonomous agent operating in the Webshop e-commerce environment.
Your task is to: Find me slip resistant, non slip men's loafers & slip-ons with rubber outsole, rubber sole with color: 1877blue, and size: 11.5, and price lower than 70.00 dollars.
You will be given the history of a past experience.
Your job is to **reflect on the past sequence**, identify any **mistakes or inefficiencies**, and then devise a **concise, improved plan** starting from the original initial state.
Below are the last few actions and corresponding observations you have:
{history_actions}
The task is NOT successfully completed.
Now it's your turn to reflect on the past experience and come up with a new plan of action.
- Your response should first be step-by-step reasoning about the strategy and path you took to attempt to complete the task. Identify where things went wrong or could be better.
- Then devise a concise, new plan of action that accounts for your mistake with reference to specific actions that you should have taken.
- Finally, end the response with your reflection and improved plan inside `<remark>` `</remark>` tags, to guide the next trial.

---

## B.4 ALFWORLD

---

**ALFWorld Standard Prompt**

You are an expert agent operating in the ALFRED Embodied Environment.
-= Welcome to TextWorld, ALFRED! =-

You are in the middle of a room. Looking quickly around you, you see a bed 1, a desk 2, a desk 1, a drawer 6, a drawer 5, a drawer 4, a drawer 3, a drawer 2, a drawer 1, a garbagecan 1, a laundryhamper 1, a safe 1, a shelf 6, a shelf 5, a shelf 4, a shelf 3, a shelf 2, and a shelf 1.

Your task is to: put a mug in desk.

{past_experience_reflection}{history_actions}
Your admissible actions of the current situation are:
'go to bed 1',
'go to desk 1',
'go to desk 2',
'go to drawer 1',
'go to drawer 2',
'go to drawer 3',
'go to drawer 4',
'go to drawer 5',
'go to drawer 6',
'go to garbagecan 1',
'go to laundryhamper 1',
'go to safe 1',
'go to shelf 1',
'go to shelf 2',
'go to shelf 3',
'go to shelf 4',
'go to shelf 5',
'go to shelf 6',
'inventory',
'look'.
Now it's your turn to take an action.
- Your response should first by step-by-step reasoning about the current situation.
- Once you've finished your reasoning, you should choose an admissible action for current step and present it within `<action> </action>` tags.

---

---

**ALFWorld Reflection Prompt**

You are an expert agent operating in the ALFRED Embodied Environment.
-= Welcome to TextWorld, ALFRED! =-

You are in the middle of a room. Looking quickly around you, you see a bed 1, a desk 2, a desk 1, a drawer 6, a drawer 5, a drawer 4, a drawer 3, a drawer 2, a drawer 1, a garbagecan 1, a laundryhamper 1, a safe 1, a shelf 6, a shelf 5, a shelf 4, a shelf 3, a shelf 2, and a shelf 1.

Your task is to: put a mug in desk.

You will be given the history of a past experience.
Your job is to **reflect on the past sequence**, identify any **mistakes or inefficiencies**, and then devise a **concise, improved plan** starting from the original initial state.

Below are the actions you took and the corresponding observations:
{history_actions}
The task is NOT successfully completed.
Now it's your turn to reflect on the past experience and come up with a new plan of action.
- Your response should first be step-by-step reasoning about the strategy and path you took to attempt to complete the task. Identify where things went wrong or could be better.
- Then devise a concise, new plan of action that accounts for your mistake with reference to specific actions that you should have taken.
- Finally, end the response with your reflection and improved plan inside `<remark></remark>` tags, to guide the next trial.

---

## C  TRAINING DETAILS

LAMER is compatible with standard policy gradient algorithms. Without specification, we use GiGPO as the default optimization algorithm. During training, the self-reflection step is also explicitly trained using the reward in the subsequent episodes. During training, we match the total number of experiences sampled for each example between RL and Meta-RL to ensure a fair comparison. Specifically, for each sample $N = 3$ episodes and set group size to 8 for Meta-RL, and use a group size of 24 for standard RL training. Besides that, other hyper-parameters and configuration are kept the same between RL and Meta-RL training. We use Qwen3-4B as the base model and train it with Adam optimizer and a learning rate of $1e - 6$. For Sokoban and MineSweeper, we train the agents with a batch size of 16 for 300 epochs. In comparison, we use batch size of 8 and 150 epochs for Webshop and ALFWorld. The environment reward is set to be 10 for successful trajectories and 0 for unsuccessful ones. We use temperature of 1.0 during rollout and 0.7 during evaluation. The maximum number of output tokens is set to 1024. Our code is based on the training framework of verl (Sheng et al., 2025) and verl-agent (Feng et al., 2025).

## D  ADDITIONAL RESULTS

### D.1  EXPERIMENTS ON DIFFERENT ARCHITECTURE

LAMER is designed as a general framework built on meta-RL principles and is model-agnostic. To validate this, we additionally ran experiments on Llama3.1-8B-Instruct (Grattafiori et al., 2024), showing that our method works well on models in a different model architecture and model size.

Table 4: Performance on Sokoban and MineSweeper environments using Llama3.1-8B-Instruct as base model. The results of p@1, p@2 and p@3 denote the success rate (%) under 1, 2, and 3 attempts, respectively.

| Method | Sokoban | | | MineSweeper | | |
|---|---|---|---|---|---|---|
| | p@1 | p@2 | p@3 | p@1 | p@2 | p@3 |
| GiGPO | 5.9 | 6.3 | 6.3 | 39.1 | 41.4 | 42.2 |
| LAMER | **17.6** | **19.9** | **20.3** | **47.7** | **61.3** | **65.6** |

In Table 4, we compare LAMER with the strongest baseline GiGPO (Feng et al., 2025), on Sokoban and MineSweeper. We show The results indicate that on Llama-3.1-8B-Instruct, LaMer still outperforms the RL baselines across the environments, demonstrating the general applicability of our method to different model architecture and size.

### D.2  COMPARISON TO RL BASELINES WITH INTER-EPISODE MEMORY

In our main experiment at Table 1, we follow previous work and evaluate the standard RL methods without access to the inter-episode memory. For comprehensive evaluation, we further evaluate the RL trained agents with access to the inter-episode memory (reflections and previous trajectories). The results of pass@3 are shown in Table 5. We observe that the inter-episode memory enhances the performance of RL trained agents on Sokoban (+3.8%) and MineSweeper (+5.3%), while degrades the performance on Webshop (-1.2%). Nevertheless, LAMER still substantially outperforms RL baselines across all the environments, demonstrating the advantage of the proposed method.

Table 5: Performance of RL baselines with access to inter-episode memory (pass@3).

| Method | Sokoban | MineSweeper | Webshop |
|---|---|---|---|
| GiGPO (w/o memory) | 44.1 | 55.1 | 75.2 |
| GiGPO (w/ memory) | 47.9 | 60.4 | 74.0 |
| LAMER | **55.9** | **74.4** | **89.1** |

# E    EXAMPLES

In Figure 6, we provide an example of trajectories and corresponding reflections produced by the agent when solving the MineSweeper game. Here each trajectory is represented by a sequence of clicks (numbered cells) on the board. The mines are not visible to the agent and will lead to failure of the game if clicked.

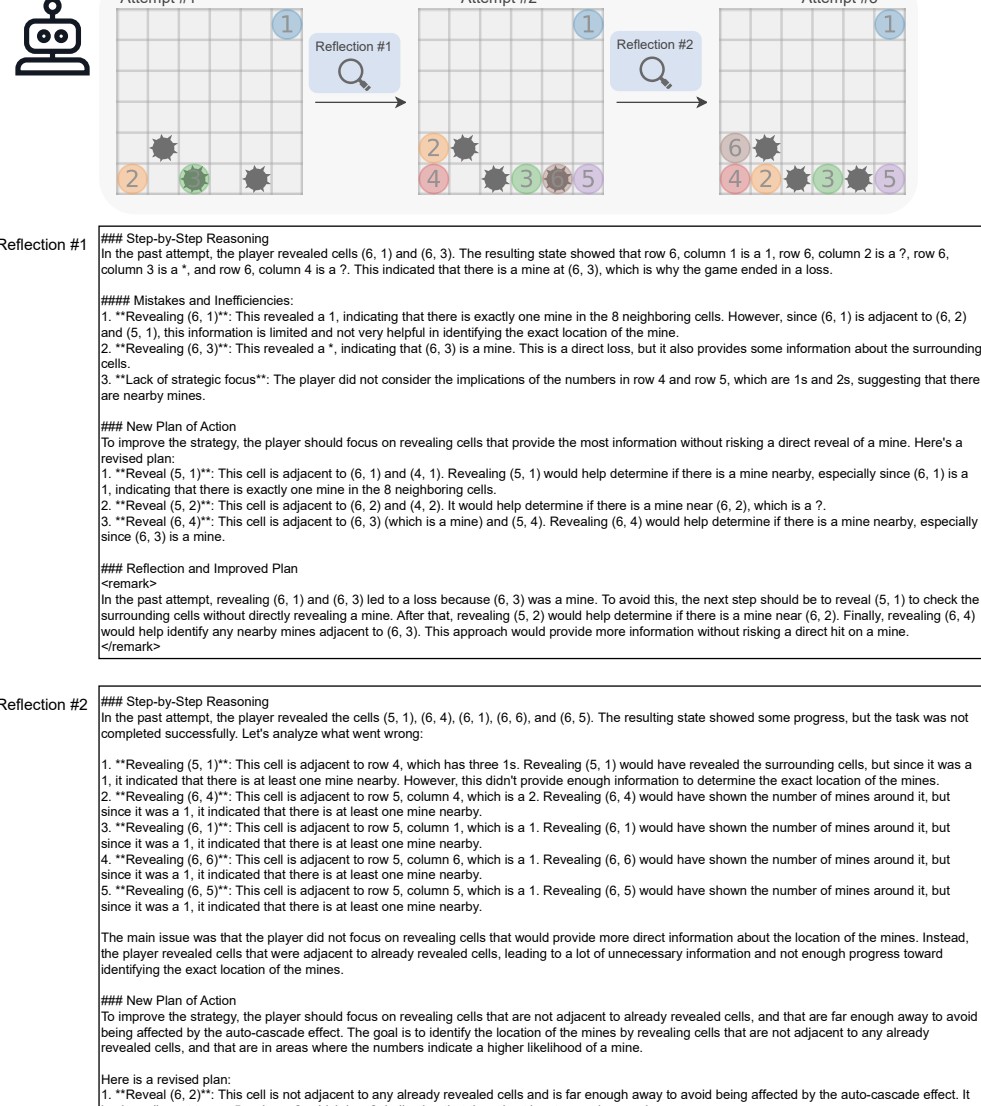

**Reflection #1**

### Step-by-Step Reasoning
In the past attempt, the player revealed cells (6, 1) and (6, 3). The resulting state showed that row 6, column 1 is a 1, row 6, column 2 is a ?, row 6, column 3 is a *, and row 6, column 4 is a ?. This indicated that there is a mine at (6, 3), which is why the game ended in a loss.

#### Mistakes and Inefficiencies:
1. **Revealing (6, 1)**: This revealed a 1, indicating that there is exactly one mine in the 8 neighboring cells. However, since (6, 1) is adjacent to (6, 2) and (5, 1), this information is limited and not very helpful in identifying the exact location of the mine.
2. **Revealing (6, 3)**: This revealed a *, indicating that (6, 3) is a mine. This is a direct loss, but it also provides some information about the surrounding cells.
3. **Lack of strategic focus**: The player did not consider the implications of the numbers in row 4 and row 5, which are 1s and 2s, suggesting that there are nearby mines.

### New Plan of Action
To improve the strategy, the player should focus on revealing cells that provide the most information without risking a direct reveal of a mine. Here's a revised plan:
1. **Reveal (5, 1)**: This cell is adjacent to (6, 1) and (4, 1). Revealing (5, 1) would help determine if there is a mine nearby, especially since (6, 1) is a 1, indicating that there is exactly one mine in the 8 neighboring cells.
2. **Reveal (5, 2)**: This cell is adjacent to (6, 2) and (4, 2). It would help determine if there is a mine near (6, 2), which is a ?.
3. **Reveal (6, 4)**: This cell is adjacent to (6, 3) (which is a mine) and (5, 4). Revealing (6, 4) would help determine if there is a mine nearby, especially since (6, 3) is a mine.

### Reflection and Improved Plan
<remark>
In the past attempt, revealing (6, 1) and (6, 3) led to a loss because (6, 3) was a mine. To avoid this, the next step should be to reveal (5, 1) to check the surrounding cells without directly revealing a mine. After that, revealing (5, 2) would help determine if there is a mine near (6, 2). Finally, revealing (6, 4) would help identify any nearby mines adjacent to (6, 3). This approach would provide more information without risking a direct hit on a mine.
</remark>

**Reflection #2**

### Step-by-Step Reasoning
In the past attempt, the player revealed the cells (5, 1), (6, 4), (6, 1), (6, 6), and (6, 5). The resulting state showed some progress, but the task was not completed successfully. Let's analyze what went wrong:

1. **Revealing (5, 1)**: This cell is adjacent to row 4, which has three 1s. Revealing (5, 1) would have revealed the surrounding cells, but since it was a 1, it indicated that there is at least one mine nearby. However, this didn't provide enough information to determine the exact location of the mines.
2. **Revealing (6, 4)**: This cell is adjacent to row 5, column 4, which is a 2. Revealing (6, 4) would have shown the number of mines around it, but since it was a 1, it indicated that there is at least one mine nearby.
3. **Revealing (6, 1)**: This cell is adjacent to row 5, column 1, which is a 1. Revealing (6, 1) would have shown the number of mines around it, but since it was a 1, it indicated that there is at least one mine nearby.
4. **Revealing (6, 6)**: This cell is adjacent to row 5, column 6, which is a 1. Revealing (6, 6) would have shown the number of mines around it, but since it was a 1, it indicated that there is at least one mine nearby.
5. **Revealing (6, 5)**: This cell is adjacent to row 5, column 5, which is a 1. Revealing (6, 5) would have shown the number of mines around it, but since it was a 1, it indicated that there is at least one mine nearby.

The main issue was that the player did not focus on revealing cells that would provide more direct information about the location of the mines. Instead, the player revealed cells that were adjacent to already revealed cells, leading to a lot of unnecessary information and not enough progress toward identifying the exact location of the mines.

### New Plan of Action
To improve the strategy, the player should focus on revealing cells that are not adjacent to already revealed cells, and that are far enough away to avoid being affected by the auto-cascade effect. The goal is to identify the location of the mines by revealing cells that are not adjacent to any already revealed cells, and that are in areas where the numbers indicate a higher likelihood of a mine.

Here is a revised plan:
1. **Reveal (6, 2)**: This cell is not adjacent to any already revealed cells and is far enough away to avoid being affected by the auto-cascade effect. It is also adjacent to row 5, column 2, which is a 2, indicating that there is at least one mine nearby.
2. **Reveal (6, 3)**: This cell is not adjacent to any already revealed cells and is far enough away to avoid being affected by the auto-cascade effect. It is also adjacent to row 5, column 3, which is a 2, indicating that there is at least one mine nearby.
3. **Reveal (6, 1)**: This cell is adjacent to row 5, column 1, which is a 1, indicating that there is at least one mine nearby. However, since it is adjacent to row 6, column 2, which is a 1, it is important to reveal this cell to confirm the location of the mine.
4. **Reveal (6, 4)**: This cell is adjacent to row 5, column 4, which is a 2, indicating that there is at least one mine nearby. However, since it is adjacent to row 6, column 3, which is a 1, it is important to reveal this cell to confirm the location of the mine.
5. **Reveal (6, 5)**: This cell is adjacent to row 5, column 5, which is a 1, indicating that there is at least one mine nearby. However, since it is adjacent to row 6, column 4, which is a 1, it is important to reveal this cell to confirm the location of the mine.

By following this plan, the player can identify the location of the mines more effectively and avoid the mistakes made in the previous attempt.
<remark>
In the past attempt, the player revealed cells that were adjacent to already revealed cells, leading to a lot of unnecessary information and not enough progress toward identifying the exact location of the mines. To improve the strategy, the player should focus on revealing cells that are not adjacent to any already revealed cells, and that are far enough away to avoid being affected by the auto-cascade effect. The revised plan involves revealing cells such as (6, 2), (6, 3), (6, 1), (6, 4), and (6, 5) to identify the location of the mines more effectively.
</remark>

Figure 6: Example of trajectories and reflections produced by LAMER trained agents on the MineSweeper game.

