# META-RL INDUCES EXPLORATION IN LANGUAGE AGENTS

## ABSTRACT

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

Building upon this, we propose LAMER (**LLM A**gent with **MetaR**L), a general MetaRL framework for LLM agent training. LAMER contains two important design factors. First, unlike standard single-episode RL, LAMER is designed around a multi-episode structure to train the agent to solve the problem through trial and error. In early episodes, the agent is encouraged to gather diverse experiences and useful information of the environment, which are used to adapt its policy in later episodes. By maximizing long-term rewards across episodes, the agent internalizes a learning algorithm that explicitly incentivizes exploration for improved downstream exploitation. Second, at both training and test time, the agent effectively leverages the feedback and reflection from the past episodes to determine the strategy for the next episode, which essentially implements an RL algorithm in context, and making the approach naturally suited for LLM agents. MetaRL produces more diverse samples while simultaneously achieving higher performance, reaching a better balance between exploration (Figure 1). To the best of our knowledge, this is the first time a meta-RL framework is used for LLM agent training.

We evaluate LAMER on four challenging long-horizon tasks: Sokoban (Racanière et al., 2017), MineSweeper (Li et al., 2024), Webshop (Yao et al., 2022) and ALFWorld (Shridhar et al., 2020). Using Qwen3-4B (Yang et al., 2025), we demonstrate that LAMER consistently outperforms prompting and RL baselines on all the environments. In addition, we observe that the trained model has learned a balance between exploration and exploitation, resulting in a better test-time scaling performance. In particular, LAMER adapts the trained policy at test time with 13% performance gains on Sokoban and more than 20% performance gain on MineSweeper and Webshop. Finally, we show that LAMER trained model achieves a better generalization to harder and out-of-distribution tasks.

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

**MetaRL obtains better performance.** Across all three environments, LAMER trained with MetaRL consistently outperforms both prompting-based baselines and RL-training methods on the

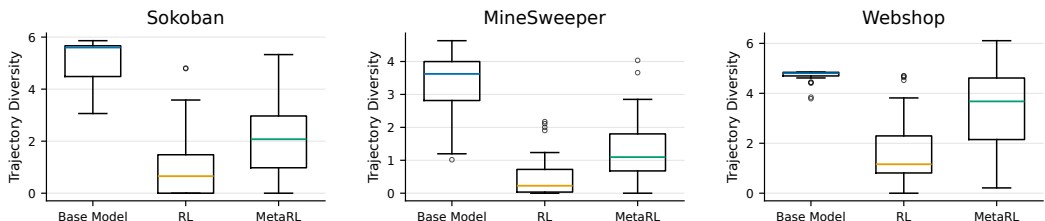

Figure 3: Trajectory diversity of base and trained models. Compared to RL, MetaRL preserves more diverse trajectories from the base model, striking a better balance between exploration and exploitation.

final pass@3 success rate. Moreover, even at the second attempt (*i.e.*, pass@2), LAMER already surpasses alternative methods. Specifically, on Sokoban, LAMER achieves a 55.9% success rate at pass@3, compared to 44.1% from the strongest RL baseline (GiGPO). In MineSweeper, MetaRL reaches 74.4% on pass@3, substantially surpassing GiGPO's result of 55.1%. On Webshop, LAMER achieves 89.1%, which is over 15% higher than the best RL-trained model. These results show substantial performance gain obtained with the LAMER framework.

**MetaRL exhibits stronger test-time scaling.** Beside achieving the best final pass@3 performance, MetaRL also demonstates test-time scaling, with larger performance gains across attempts according to the results in Tabel 1. For example, the improvement from pass@1 to pass@3 is 23.5% in MetaRL, significantly larger both RL-trained and prompting-based baselines (which are less than 5%). Notably, although MetaRL starts with slightly lower pass@1 performance than GiGPO in MineSweeper and Webshop, it quickly recovers and surpasses all baselines by pass@2 and pass@3. The results indicate that the trained model has successfully learned to adapt effectively from prior mistakes, leading to significant gains in subsequent attempts.

**MetaRL induces exploration.** To further analyze the behavior of the models, we measure the diversity of answer trajectories across environments. For each question, we sample multiple trajectories from the agent and group the identical trajectories that have the same states and actions. These groups are used to form the empirical distribution over distinct trajectories, as shown in Figure 1. We then estimate the entropy the distribution to quantify the *trajectory diversity*. Figure 3 compares the trajectory diversity of the base model, RL, and MetaRL agents across environments. We observe that the base model exhibits the highest entropy, indicating it generates a wide range of trajectories, though this diversity does not translate into higher success rates (see Table 1). RL trained agent reduces diversity and converges toward more deterministic behaviors. In contrast, MetaRL preserves a higher level of diversity than RL, allowing more exploration at test time.

## 5.3 INFLUENCE OF TRAJECTORY DISCOUNT FACTOR

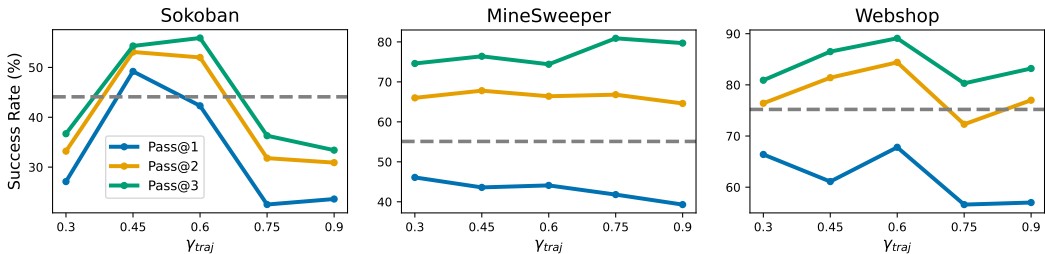

Figure 4: Success rates of models trained with different $\gamma_{\text{traj}}$. A higher value encourages more exploration during training.

Discount factor $\gamma_{traj}$ controls the cross-episodes rewards propagation and it balances between exploration and exploitation in the LAMER framework. We further study its influence on the per-

formance across different environments. Figure 4 shows the results of the LAMER models trained with different trajectories discount factors. We observe that each environment will favor different discount factors. For Sokoban and Webshop, intermediate values like 0.6 yield the best results, suggesting that balancing immediate and long-term rewards is more important for these tasks. In contrast, MineSweeper benefits from relatively larger $\gamma_{traj}$ like 0.9, indicating that extended credit assignment better supports strategic exploration in this environment. Overall, the results show that $\gamma_{traj}$ provides a practical way to control the trade-off between exploration and exploitation across environments.

## 5.4 GENERALIZATION TO HARDER TASKS

Next we study the generalization ability of the pretrained models on harder tasks. To this end, we take the models trained with RL and MetaRL and evaluate them on the harder tasks in the environments of Sokoban and MineSweeper. We increase the difficulty by using more boxes for Sokoban and more mines for MineSweeper in the grid. The results are shown in Figure 5. As expected, the model trained with both RL and MetaRL underperforms on harder tasks with increasing number boxes or mines in the grid. However, MetaRL consistently outperform RL on all the difficulty levels. Notably, on the most difficult setting, the model trained from MetaRL still outperforms the RL trained model with 10% performance gap on Sokoban, and 5% performance gap on the MineSweeper. The consistent gap indicates that LAMER trained with MetaRL not only performs better on the training distribution, but also generalizes better to the harder tasks.

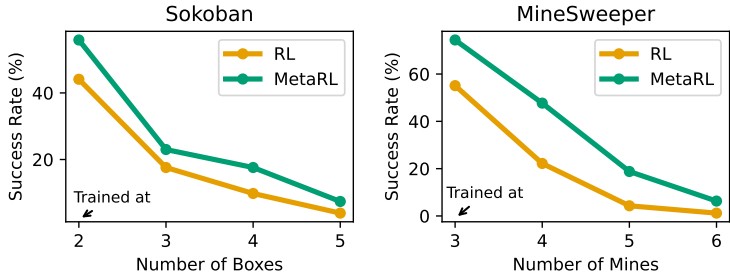

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

### LLM USAGE STATEMENT

LLM is mainly used for proofreading and as a plot assistant in this work.

### REPRODUCIBILITY STATEMENT

In order to ensure that our work is reproducible, we have provided experimental details in Section 5.1, together with the template of prompts we used in Appendix B. Complete code documentation is under development and will be made available alongside the paper's final version.