# OpenReview forum: "Meta-RL Induces Exploration in Language Agents"
_ICLR.cc/2026/Conference — ICLR 2026 Poster_

### Official Review · Reviewer_Qdy5 · 2025-10-18

**Soundness:** 2
**Presentation:** 3
**Contribution:** 2
**Rating:** 6
**Confidence:** 3

**Summary:**

This paper proposes a meta-reinforcement learning framework, LaMer, which is designed to induce exploration and adaptation in large language model agents. Unlike conventional single-episode RL setups, LaMer uses a cross-episode training paradigm, where an agent interacts with the same environment across multiple episodes, balancing exploration and exploitation through a trajectory discount factor. The method leverages in-context policy adaptation via reflection, enabling agents to improve within trials without parameter updates.

The authors evaluate the method on Sokoban, MineSweeper, WebShop, and ALFWorld show that LaMer outperforms prompting-based and RL baselines (PPO, GRPO, GiGPO). It achieves a good marginal performance gain on complex environments and improved generalization to harder or out-of-distribution tasks. The framework introduces a bridge between meta-learning and RL for LLM-based agents.

**Strengths:**

- Novel framework design: The cross-episode MetaRL formulation extends standard RL to multi-episode exploration, introducing a new axis of “meta-time” learning that aligns well with LLMs’ in-context adaptation capabilities. The introduction of the trajectory discount factor provides interpretable control over the exploration–exploitation tradeoff.

- Improvements over the current baselines over benchmarks: LaMer consistently outperforms RL baselines across all benchmarks (Table 1), achieving +20% absolute improvement on Minesweeper and WebShop. Figures 3–5 show substantial increases in trajectory diversity and test-time scaling behavior, supporting the exploration hypothesis.

- Good generalization evidence: Demonstrated robustness to task difficulty (Figure 5) and generalization to unseen ALFWorld tasks (Table 2). Shows that meta-learned exploration strategies can generalize beyond the training distribution, which is an important step toward adaptive, open-ended agents.

- Broader impact: Provides a reasonable path to unify RL-based training and in-context learning for future meta-learning language agents. Bridges recent work in test-time compute scaling and reasoning through reflection.

**Weaknesses:**

- Limited novelty in algorithmic components: While the formulation is clean, most components, cross-episode return accumulation and reflection-based adaptation, are incremental combinations of known techniques. The paper could better articulate what is fundamentally new beyond adapting Meta-RL for LLMs.

- Insufficient analysis of failure cases: The paper lacks qualitative examples or ablation studies on when reflection adaptation fails (e.g., misleading feedback loops or hallucinated self-reflection). As well, it does not have analysis of how context length or reflection prompt quality impacts learning efficiency.

- Sequential dependency and compute inefficiency: Section 5.6 notes that MetaRL training is 2–3× slower due to sequential episodes, but does not propose mitigation strategies or quantify the trade-off precisely (e.g., wall-clock vs. sample efficiency).

- Evaluation fairness concerns: It is unclear whether RL baselines had access to equivalent multi-episode feedback or if they were constrained to single-episode objectives, potentially overstating MetaRL’s advantage.

**Questions:**

- Meta-objective design: How sensitive is performance to the trajectory discount factor? Could adaptive scheduling of this factor improve learning stability?

- Reflection mechanism: Did the authors experiment with alternative reflection prompt formats or automatic summarization of past episodes? It would be great to add it in the ablations.

- Training efficiency: Can rollout dependencies be partially parallelized (e.g., via curriculum or pseudo-batch episodes) without breaking credit assignment? It will result higher training efficiency.

- Exploration quality metrics: Besides trajectory diversity entropy, are there other quantitative metrics used to measure exploration effectiveness?

---

> ### Author Response · Authors · 2025-11-22
> **Rebuttal [1/2]**
>
> We thank the reviewer for the valuable feedback. We are delighted that the reviewer finds our framework design novel, method effective with good generalization and recognizes the broader impact of our work. We address the reviewer’s questions and concerns below.
>
> ---
>
> **`W1: While the formulation is clean, there is limited Novelty in algorithmic components.`**
>
> We thank the reviewer for the important question. While LaMer is inspired by the prior works, our main contribution is to introduce a **conceptual shift on how LLM agents should be trained**.
>
> In particular, existing RL-based methods are designed to optimize the single-episode returns, which encourages deterministic policies and limits their ability to explore or adapt to new environments. Our work instead formulates LLM agent training as a meta-reinforcement learning problem across multiple episodes. This paradigm shift is crucial because it explicitly trains the agent to explore and self-adapt, moving beyond the simple prompting or imitation approaches in the previous works.
>
> As a result, the proposed Meta-RL framework fundamentally changes the agent learning process:
> - It incentivizes **exploratory behavior** which is limited in standard RL training.
> - It enables **effective in-context adaptation at test time**, resulting in strong test time scaling.
> - It improves **generalization to harder and out-of-distribution tasks**, where existing methods are limited.
>
> Thus, the core contribution of LaMer lies in reframing LLM agent training around cross-episode learning and adaptation, leading to qualitatively new capabilities that previous RL or prompting methods do not produce.
>
> ---
>
> **`W2: Insufficient analysis of failure cases: any qualitative examples (e.g., misleading feedback loops or hallucinated self-reflection)?`**
>
> Thank you for the suggestion. We’ve observed that the most frequent failure case is *ineffective reflection*, where the model generates reflections that are overly generic and provide little actionable guidance for the next episode. For example, on Sokoban, sometimes the model will generate reflection such as:
>
> > *To ensure all boxes are properly placed onto targets, we need to systematically address each box and target. The initial plan didn't account for the need to push each box onto its specific target, leading to incomplete placement… *
>
> **Such reflection is a high-level statement and does not provide meaningful information** to improve the subsequent episode. In comparison, reflection that contains a clear plan of actions in the next episode tends to be much more effective. We will include discussion and representative examples of failure cases in the revision of manuscript.
>
> ---
>
> **`W3 and Q3: Can rollout dependencies be partially parallelized (e.g., via curriculum or pseudo-batch episodes) without breaking credit assignment?`**
>
> Thank you for the great suggestion. Our current implementation of Meta-RL rollouts is intentionally kept simple to facilitate algorithm development, and we agree that there’re several aspects to explore for improving the training efficiency:
>
> 1. **Asynchronous rollout**. Our current implementation synchronizes workers at every environment step, which leads to significant idle time and wasted compute. Recent RL frameworks (e.g., veRL, slime, RLLM) start to support asynchronous rollouts. We expect that incorporating such async design could greatly improve throughput.
> 2. **Curriculum over episode count**. As you suggested, we can begin training with a small number of episodes, and then increase the episode over training, yielding the benefits of Meta-RL while reducing early-stage overhead.
> 3. **Trajectory caching / pseudo-batch episodes**. In line with the idea of pseudo-batching, we can cache the first exploratory trajectory from the previous epoch and reuse it to warm-start the next trial.
>
> We note that the throughput is also influenced by the hardware constraints, i.e.,  the maximum number of sequences that can be processed concurrently. When this limit is smaller than the rollout size, even RL rollout can also not be fully parallelized, making its efficiency closer to Meta-RL. We will update our discussion in the manuscript.

---

> > ### Author Response · Authors · 2025-11-22
> > **Rebuttal [2/2]**
> >
> > **`W4: Comparison RL baselines that are given access to equivalent multi-episode feedback.`**
> >
> > Thank you for the important question. In the manuscript, the RL-baselines are evaluated without inter-episode memory, consistent with the standard practice. Following the reviewer’s suggestion, we further evaluate RL baselines with *inter-episode memory*, allowing them to leverage the history across the episodes. The results are shown below:
> >
> > *Table 1:  Evaluation of RL baseline with and without inter-episode memory*
> > |                 | Sokoban | MineSweeper | Webshop |
> > |-----------------|---------|-------------|---------|
> > | RL (w/o memory) |  44.1   |   55.1      |  75.2   |
> > | RL (w/ memory)  |  47.9   |   60.4      |  74.0   |
> > | Meta-RL         | **55.9** |  **74.4**  | **89.1**|
> >
> >
> > We observe that the inter-episode memory enhances the performance of RL trained agents on Sokoban (+3.8%) and MineSweeper (+5.3%), while degrades the performance on Webshop (-1.2%).
> > Nevertheless, **Meta-RL still substantially outperforms RL baselines across all the environments**, demonstrating the advantage of the proposed method. We’ll incorporate the discussion in the revised manuscript.
> >
> > ---
> >
> > **`Q1: How sensitive is performance to the trajectory discount factor? Could adaptive scheduling of this factor improve learning stability?`**
> >
> > Thank you for the insightful question. We investigated the sensitivity of LaMer to the trajectory discount factor $\gamma_{traj}$ in Section 5.3 (Figure 4) of the manuscript. For clarity, we summarize the results below:
> >
> > *Table 2: Performance of LaMer with different $\gamma_{traj}$ and the RL Baseline*
> > |             | 0.3  | 0.45 | 0.6      | 0.75     | 0.9  | Baseline |
> > |-------------|------|------|----------|----------|------|----------|
> > | Sokoban     | 36.7 | 54.3 | **55.9** | 36.3     | 33.4 | 44.1     |
> > | MineSweeper | 74.6 | 76.4 | 74.4     | **80.9** | 79.7 | 55.1     |
> > | Webshop     | 80.9 | 86.5 | **89.1** | 80.3     | 83.2 | 75.2     |
> >
> > We observe that
> > - LaMer consistently outperforms the baseline across most choices of $\gamma_{traj}$ (0.45 and 0.6 on Sokoban, and all tested values on MineSweeper and Webshop).
> > - The optimal $\gamma_{traj}$ varies across environments, reflecting different exploration–exploitation trade-offs.
> >
> > We highlight that **a fixed value of 0.6 already provides stable and competitive performance** across all the environments. We also agree that adaptive scheduling could potentially yield further improvements, and we will include additional discussion in the revised manuscript.
> >
> > ---
> >
> > **`Q2 and W2: Alternative reflection prompt formats or automatic summarization of past episodes?`**
> >
> > We thank the reviewer for the great suggestions. Our current prompt contains both history trajectories and corresponding reflections. Following the reviewer’s suggestion, we conduct the ablation of prompts and compare the results of LaMer with (1) both reflection and history (reflection + history), (2) only the self-reflection (reflection_only) and (3) only the previous trajectories (history_only). We summarize the results below.
> >
> > *Table 3: Performance of LaMer with different context configurations*
> > |                            | Sokoban | MineSweeper | Webshop |
> > |----------------------------|---------|-------------|---------|
> > | LaMer (Reflection+history) |  55.9   |   74.4      |  89.1   |
> > | LaMer (Reflection-only)    |  **56.4**   |   **80.5**      |  **92.8**   |
> > | LaMer (History-only)       |  34.8   |   69.5      |  89.3   |
> >
> >
> > The results indicate that
> > - (1st or 2nd row) Self-reflection provides clear benefit over using only the history trajectories.
> > - (2nd row) It’s also interesting to see that “reflection-only” outperforms “reflection + history” across all environments. We hypothesize “reflection-only” offers a more concise and clear guidance in the context, leading to better adaptation of the agent’s behavior.
> >
> > We’ll include the results and discussion in the revised manuscript.
> >
> > ---
> >
> > **`Q3: Are there other quantitative metrics used to measure exploration effectiveness?`**
> >
> > That’s a great question. We believe that the “effectiveness” of exploration depends on whether the trajectory reveals important information about the environment that could be leveraged in the following attempt. Therefore, the performance improvement between attempts (pass@3 - pass@1) can also serve as a metric on the effectiveness of exploration. Based on this, we summarize the results below
> >
> > *Table 4: Comparison of the performance gain across attempts*
> > |         | Sokoban | MineSweeper | Webshop |
> > |---------|---------|-------------|---------|
> > | GiGPO   | 2.5     | 3.1         | 1.8     |
> > | LaMer   | 13.5    | 30.3        | 21.3    |
> >
> >
> > The results show that LaMer trained agents obtain significantly larger improvement between attempts compared to RL baselines, suggesting that their exploratory trajectories might be more informative and effective.

---

> > > ### Comment · Reviewer_Qdy5 · 2025-11-23
> > >
> > > Thanks for the detailed response. I'll keep my score.

---

### Official Review · Reviewer_xsGo · 2025-10-22

**Soundness:** 3
**Presentation:** 3
**Contribution:** 3
**Rating:** 6
**Confidence:** 4

**Summary:**

The paper proposes a Meta-RL framework for LLM agents, combining cross-episode RL and in-context reflection. The idea is to train the model to learn to explore by first gathering information, then exploiting it across the following episodes. Results show clear gains over RL and prompting baselines across Sokoban, MineSweeper, Webshop, and ALFWorld. The approach seems effective at leveraging multi-episode structure and reflection to adapt at test time, though it’s unclear whether the gains truly stem from “learning to explore” or from other confounding factors such as longer contextual horizons or additional information flow between episodes.

**Strengths:**

The idea is clear and well motivated. It’s conceptually simple and general. The evaluation is broad and the study of generalization is interesting. The paper is clear and well-written.

**Weaknesses:**

**What is actually learned?**
It’s not so clear to me whether the agent is only conditioned on the reflection from the previous episode or on the previous history too? If conditioned on both, it would be good to ablate the reflection mechanism and see how performance holds. Ablating history and only keeping reflections might be interesting too.

Is the feedback generation capacity trained too? Or just leveraging a frozen model? Maybe the approach doesn’t actually train exploratory behaviors but only improves feedback generation and feedback use in the following episode.

To make sure it’s not the case, it would be interesting to visualize exploration strategies in the first episode: what are agents doing? Maybe through videos on a website. Looking and analysing generated feedback might be useful too. Overall I would like to know more about the exploratory behaviors this paper claims is being trained.

It might be interesting too to feed a first episode collected by either 1) random actions, or 2) the base model; then generate feedback and run the second episode with the trained Meta-RL agent. Is there a drop in performance? If so, it means the meta-RL agent truly learned to explore. If not, then exploration doesn’t matter so much, what matters is the feedback offering a “second chance”.

Are the different episodes in a group the same task instance? e.g. with mines in the same spot in mine sweeper? I assume it’s the case, otherwise exploration would be useless, right? In that case, in Sokoban and MineSweeper, it may not be so much about “learning to explore” as it is about giving a second chance the agent can actually leverage (second chances can’t be leveraged in the standard RL condition because there is no inter-episode memory).



**Missing information about training task distribution**
It’s unclear how different training tasks are from each other. If they’re similar, “learning to explore” might just mean reusing the same strategy in all task instances? Are agents adapting their exploration (1st episode) to the task instance? Here again looking at replays would be useful.

**Comparisons**
The RL baseline controls for total experience but doesn’t control for the length of experience the agent has access to at any given moment. One way to control for this would be to run the RL agents on longer episodes instead of running it on longer trajectories.

**Generalization to harder tasks**
I don’t know if you can say that meta-RL generalizes better here: performance drops by similar percentages in both cases. Meta-RL doesn’t “hold better”, it just starts higher. This is a bit of an overclaim. The AlfWorld generalization study is more convincing.


**Typos:**
* fari comparison -> fair
* Tabel 1 -> Table 1

**Questions:**

* How is the “empirical distribution over distinct trajectories” computed exactly?
* Are prompting baselines also given access to memories across episodes? Reflection only? or histories too?
* In addition to “bold” indicating best performance, it is useful to make other performance that are not statistically significantly worse than the best (e.g. underlined).

---

> ### Author Response · Authors · 2025-11-22
> **Rebuttal [1/2]**
>
> We thank the Reviewer for the valuable feedback and a positive evaluation of our work. We are encouraged that the reviewer finds the paper clear and well written. We address the reviewer’s questions and concerns below.
>
> ---
>
> **`W1: What is learned by the agent? Is the feedback generation capacity trained? Or just leveraging a frozen model?`**
>
> In our Meta-RL framework, the agent *jointly* learns the ability of **exploration** and **self-reflection** through the multi-episode training. The self-reflection step is *not* frozen but is trained using the reward from the subsequent episode. The joint reinforcement of both abilities are crucial to the success of our framework, resulting in more diverse trajectories (Figure 3) and better test-time scaling (Table 1 in the paper) on the trained agents. We include additional explanations in the revised paper.
>
> ---
>
> **`W2: What is the agent conditioned on? + Ablation on the reflection mechanism (e.g. only keeping reflections or history).`**
>
> At each step, the agent is conditioned on the accumulated context, including both reflection and trajectory history from previous attempts. Following reviewer’s suggestion, we conduct the ablation of training agents with different content in the agent memory (reflection + history vs. reflection only vs. history only ). The results are summarized below.
>
> *Table 1: Performance of LaMer with different context configurations*
> |                            | Sokoban | MineSweeper | Webshop |
> |----------------------------|---------|-------------|---------|
> | LaMer (Reflection+history) |  55.9   |   74.4      |  89.1   |
> | LaMer (Reflection-only)    |  **56.4**   |   **80.5**      |  **92.8**   |
> | LaMer (History-only)       |  34.8   |   69.5      |  89.3   |
>
> The results show that:
> - (1st or 2nd row) Self-reflection provides clear benefit over using only the history trajectories
> - (2nd row) Interestingly, “reflection-only” outperforms “reflection + history” across all the environments. We hypothesize that “reflection-only” offers a more concise and clear guidance in the context, leading to better adaptation of the agent’s behavior.
>
> We’ll include the results and discussion in the revised manuscript.
>
> ---
>
> **`W3: Comparison on the performance of second episode from Meta-RL agent conditioned on a first episode collected by either 1) random actions, or 2) the base model.`**
>
> This is indeed a great way to gauge whether exploration is learned, we thank the reviewer for the insightful suggestion. We take the trained model and evaluate its 2nd episode performance where the 1st episode is generated 1) random actions, 2) the base model, and 3) the trained model. The results are shown in the following table.
>
> *Table 2: Performance of the second episode with varying source of the first trajectory*
> |                           | Sokoban | MineSweeper | Webshop |
> |---------------------------|---------|-------------|---------|
> | 1st Random + 2nd trained  |  37.5   |   51.6      |   48.4  |
> | ~~1st Base   + 2nd trained~~  |  ~~45.3~~   |   ~~69.5~~     |~~83.6~~ |
> | 1st Base + 2nd trained    | 31.3    | 50.0        | 53.1    |
> | 1st trained + 2nd trained | **46.9** | **68.8**   | **82.0**   |
>
> The results show that
> - (3rd row) The first episode generated by the trained model leads to better performance on the second episode.
> - (1st and 2nd row) In comparison, using the base model or random policy for the first episode will lead to a performance drop (on Sokoban and MineSweeper).
>
> This confirms that the Meta-RL trained agent has learned to conduct effective exploration in the early episodes, which is crucial for the subsequent episodes. We will include these results in the revised version of the manuscript.
>
> ---
> **`W4: Visualization of trajectories and reflections by the agent.`**
>
> We thank the reviewer for the great suggestion. We have updated the manuscript in Appendix (Figure 6) to show the trajecories and reflections produced by the trained agents.

---

> ### Author Response · Authors · 2025-11-22
> **Rebuttal [2/2]**
>
> **`W5: Second chances can’t be leveraged in the standard RL condition because there is no inter-episode memory.`**
>
> Thank you for the interesting question. In our main experiment, we follow previous work and evaluate the standard RL methods without access to the inter-episode memory. For comprehensive evaluation, we further evaluate the RL trained agents with access to the inter-episode memory. The results are shown below.
>
> *Table 3: Performance of RL baselines with and without inter-episode memory*
> |                 | Sokoban | MineSweeper | Webshop |
> |-----------------|---------|-------------|---------|
> | RL (w/o memory) |  44.1   |   55.1      |  75.2   |
> | RL (w/ memory)  |  47.9   |   60.4      |  74.0   |
> | Meta-RL         | **55.9** |  **74.4**  | **89.1**|
>
> We observe that the inter-episode memory enhances the performance of RL trained agents on Sokoban (+3.8%) and MineSweeper (+5.3%), while degrades the performance on Webshop (-1.2%).
>
> Nevertheless, **Meta-RL still substantially outperforms RL baselines across all the environments**, demonstrating the advantage of the proposed method. We’ll incorporate these results and discussion in the revised manuscript and we thank the Reviewer for suggesting it.
>
> ---
>
> **`W6: Information about training task distribution.`**
>
> We provide a description on the task distribution of each environment below:
> - **Sokoban**: Games are programmatically synthesized with randomized configurations of walls, box placements, and target layouts, determined by the random seed.
> - **Minesweeper**: Each game is generated with random mine placements and a randomized first-click position, ensuring diversity across training instances.
> - **Webshop**: Tasks are curated from a realistic online shopping website. Tasks vary on product sets, descriptions, budgets, and item ranks on the search engine.
> - **ALFWorld**: Tasks vary on the combination of {task-type, object, receptacle, room}, spanning 6 different task types and 120 different rooms.
>
> Thus, the training task distribution contains substantial variability that cannot be solved by reusing the same strategy. We’ll add the discussion in the revised manuscript.
>
> ---
>
> **`W7: Comparisons between RL and MetaRL, why not running RL for longer trajectories instead of longer episodes?`**
>
> In our experiments, we ensure fairness by matching the **total number of trajectories** and the **maximum number of environment steps** between RL and Meta-RL. We agree that one could potentially run RL agents with longer episodes instead of long trajectories. However, doing so would reduce the amount of **reward signal** the agent could receive during training, especially in environments with sparse reward. For this reason, we chose to match the number of total experiences instead.
>
> ---
>
> **`Q1: How is the “empirical distribution over distinct trajectories” computed exactly?`**
>
> For each task, we sample multiple trajectories (N=128) from the agent. We then group identical trajectories together that have the same states and actions. With this, we obtain the empirical frequency of each distinct trajectory, for which compute the entropy to quantify trajectory diversity. In response to the reviewer's question, we have updated the manuscript on Figure 1 (right). We provide a new visualization that shows a complementary visual example on a Minesweeper instance with the empirical probabilities of trajectories, to better illusrate this process.
>
> ---
>
> **`Q2: Are prompting baselines also given access to memories across episodes? Reflection only? or histories too?`**
>
> We follow the setup used in the previous work where Reflection has access to the cross-episode memories (including reflection and history) while ReAct does not.
>
> ---
>
> **`Q3: Use underline for the results that are not statistically significantly worse than the best.`**
>
> Thank you for the great suggestion! We’ll annotate results that are not statistically significantly worse from the best-performing method in the updated manuscript.

---

> > ### Comment · Reviewer_xsGo · 2025-11-24
> >
> > Thank you for the detailed answer and running these following experiments, which I believe are valuable additions to the paper.
> >
> > **About statistical significance:**
> > "We’ll annotate results that are not statistically significantly worse from the best-performing method in the updated manuscript."
> > I do not see the annotation in the new version, does it mean they haven't been added, or that every best result is statistically significantly better than the second bests?
> >
> > **Remaining concern about the "we learn to explore" claim"
> > The second new table shows that the "learned exploration" is only beneficial in Sokoban and Minesweeper, not in webshop (row 2 vs 3). Even when beneficial, it's not so clear whether these differences are significant, they are not that large. So it seems that most of the gain comes from observing the reflection on previous episodes, that could be generated by an untrained base model. In this case, the approach is not so much about "learning to explore" and more about "learning to leverage feedback from previous lives"
> >
> > One way to further support the "learning to explore" would be to look at the comparison between:
> > * episodes 1+2 with base model, 3 with trained model and
> > * episodes 1+2 with trained model, 3 with trained model
> > adding the extra episode compared to the above table, might make the difference more significative and better support the claim that the approach is learning to explore.
> >
> > It's not so clear whether these domains are well suited to demonstrate a "learning to explore" behavior. I don't much about Webshop, but MineSweeper and Sokoban don't really call for active exploration strategies: MineSweeper is about making mistakes and reflecting about them so they're not made in the following episodes: it's not so much about "learning an exploratory behavior" as it is about "giving an extra life". The same is true in Sokoban?
> >
> > The claim would be easier to support in domains that really call for an active exploration of the environment: where you could look at a trajectory and really see whether the agent is trying to "explore the environment" or "solve the environment". Maybe VGDL games? https://github.com/rubenvereecken/py-vgdl
> >
> > Thanks for adding an example. I'm not sure I understand what the numbers are about: it seems they are the order of the action sequence, but then we don't see the number giving the cue about how many bombs are around. Color heat could code for order in the sequence, and the number for the cue.
> >
> > I think the paper would benefit from showing more examples, from different domains as well. It would be nice to be able to compare these "exploratory behaviors" of the first episode with a base model vs the trained model, so we could get a better feel about the "learning to explore" claim. One easy way to scale that is to generate gifs and upload them somewhere.

---

> ### Author Response · Authors · 2025-11-27
>
> We sincerely thank the reviewer for the constructive feedback and detailed engagement, which is very helpful for improving our work.
>
> ---
>
> **`Results regarding the ablation study on "learned exploration"`**
>
> We thank the reviewer for suggesting the insightful ablation. In our initial response, the result reported for was mistakenly taken from another run of “1st trained + 2nd trained”. We apologize for the mistake and present the correct results below.
>
> *Table 1: Performance of the 2nd episode with varying sources of the 1st trajectory*
> |                           | Sokoban | MineSweeper | Webshop |
> |---------------------------|---------|-------------|---------|
> | 1st Random + 2nd trained  | 37.5    | 51.6        | 48.4    |
> | 1st Base + 2nd trained    | 31.3    | 50.0        | 53.1    |
> | 1st Trained + 2nd trained | **46.9** | **68.8**   | **82.0**  |
>
> We observe that [1st Base + 2nd trained] (2nd row) obtains similar performance to [1st Random + 2nd trained] (1st row). In comparison, [1st trained + 2nd trained] (3rd row) yields significantly better performance across all the environments, which demonstrates the effectiveness of exploration in the 1st trajectory from the trained agents.
>
> Following the reviewer’s suggestion, we further conduct the experiments to evaluate the performance on the 3rd trajectory, and the result is summarized below:
>
> *Table 2: Performance of the 3rd episode with varying sources of the 1st and 2nd trajectories*
> |                               | Sokoban | MineSweeper | Webshop |
> |-------------------------------|---------|-------------|---------|
> | 1st/2nd Random + 3rd trained  | 38.3    | 55.4        | 37.5    |
> | 1st/2nd Base   + 3rd trained  | 34.4    | 45.3        | 43.8    |
> | 1st/2nd Trained + 3rd trained | **48.4**  | **80.5**    | **90.6**  |
>
> The results show a larger performance gap on the 3rd trajectory in MineSweeper and Webshop. This further indicates the effectiveness of explorative trajectories generated by the trained agents.
>
> ---
>
> **`It's not so clear whether these domains are well suited to demonstrate a "learning to explore" behavior.`**
>
> We appreciate the reviewer’s thoughtful question about whether these environments are appropriate for studying learned exploration.  We argue that **exploration plays an essential role at both training and test time**, and this is reflected across all three environments.
>
> **During training**, an explorative model generates more diverse samples, leading to richer learning signals and better credit assignment. As shown in Figure 3, Meta-RL produces more diverse trajectories compared to the RL baselines, which will contribute to the better *training dynamics* across all the environments. This aligns with recent work on diversity-driven exploration in LLM reasoning tasks [1–4].
>
> **At test time**, exploration manifests differently depending on the environment:
> - **Sokoban (fully observable)**: Exploration resembles strategic trial-and-error. The agent examines the plausible trajectories in the solution space and identifies the errors for later episodes.
> - **MineSweeper and Webshop (partially observable)**: Exploration involves active *information gathering*, such as revealing cells or collecting search results, which improves performance in subsequent episodes.
>
> Following the reviewer’s suggestion, we visualize the 1st episodes of the base and trained agents on Sokoban and MineSweeper (5 examples each).
>
> **Visualizations of the trajectories**: https://imgur.com/a/aK0sbcG
>
> Our observations are:
> - In Minesweeper, the trained agents tend to select distant cells at corners to uncover more information about the board. In contrast, the base model often acts locally, choosing cells close to the previous actions.
> - In Sokoban, the trained agents demonstrate better performance on problem-solving. It attempts to solve the problem directly and provide valuable experiences of error to improve the subsequent episodes.
>
> These behaviors are also reflected in the performance of trained agents (Table 1 in the paper): in MineSweeper and Webshop, the agent sacrifices performance in the first episode (for information gathering) but achieves much stronger performance in later episodes.
>
> **Reference**
>
> [1] The Entropy Mechanism of Reinforcement Learning for Reasoning Language Models. arXiv: 2505.22617
>
> [2] Diversity-Aware Policy Optimization for Large Language Model Reasoning. NeurIPS, 2025
>
> [3] Reasoning with exploration: An entropy perspective. arXiv:2506.14758
>
> [4] Pass@k Training for Adaptively Balancing Exploration and Exploitation of Large Reasoning Models. arXiv:2508.10751

---

> ### Author Response · Authors · 2025-11-27
>
> **`Statistical significance of the results.`**
>
> Following the reviewer’s suggestion, we have run the results of our method and the strongest baseline (GiGPO) with 3 random seeds. We report the results of mean ± standard deviation on the table below. We annotate the best with **bold** and the results that are not significantly worse (within mean ± std from others) with *italic* (only Sokoban p@1) in the table below. However, using three seeds, which is a standard practice in the field due to computational costs, limits the power of the statistical test, as one would need many more samples.
>
> *Table 3: Performance of GiGPO and LaMer*
> |       | Sokoban p@1 | p@2        | p@3        | MineSweepr p@1 | p@2        | p@3        | Webshop p@1 | p@2        | p@3        |
> |-------|---------------|--------------|--------------|-----------------|--------------|--------------|--------------|--------------|--------------|
> | GiGPO | *36.9±3.4*    | 39.1±3.4     | 39.6±3.4     | **48.4±3.4**    | 51.9±2.9     | 52.3±2.5     | **71.9±1.4** | 72.6±1.6     | 72.8±1.8     |
> | LaMer | **37.6±3.5**  | **45.9±4.3** | **49.6±4.4** | 43.9±0.2        | **68.2±1.7** | **76.6±1.8** | 68.6±1.8     | **84.8±1.8** | **91.8±2.0** |
>
>
> ---
>
> Finally, we thank the reviewer again for the valuable and insightful comments. We will carefully take all of the feedback into account to further improve the clarity and quality of the manuscript.

---

> > ### Comment · Reviewer_xsGo · 2025-11-27
> >
> > Thank you for your engagement in this discussion
> >
> > I think I'm still not entirely convinced that these experiments show "learning to explore" vs "learning to try the task" in the first episodes. In the sense that the agent is still rewarded to solve the game in the earlier episodes if it could right? So it would actually want to solve the game from episode 1. Sometimes it does and that's great, sometimes it doesn't and episode 2 can learn from the feedback. This doesn't incentivize directly for exploring in earlier episodes.
> >
> > For me, there is two ways this setting could let the agent "learn to explore"
> >
> > 1. If the agent is not rewarded to solve the game in early episodes more than in later ones? In that way, there would be no drive to solve episode 1, and this incentivize the agent to purely explore at the beginning?
> >
> > 2. If the domains required active exploration such that they have very little chance of being solved in the first episode, and trying to solve the first episode would lead to insufficient feedback to increase the chances of solving the second episode: then the agent would learn to explore in episode 1.
> >
> > I think ideally you'd need both to show "learning to explore".
> >
> > But as far as I understand your current setting doesn't tick any of these, so it's hard to really prove that there is learning to explore:
> >
> > 1. The agent is rewarded to solve games in earlier episodes if it can: which would cause them to learn to solve the game more than learn to explore
> > 2. Trying to solve the domains would bring good enough feedback to really increase the chances of solving the game at episode 2 if episode 1 failed. In Sokoban everything is observable, so the exploration of the level is not really necessar: learning to explore=trying to solve the game. In MineSweeper there is partial observability, and I guess the agent could take "more or less risks" in the game, where "learning to explore" would then mean "learning to take more risks in the first episode".
> >
> > ---
> >
> > I think there is an experiment that would help us prove or disprove the "learning to explore" claim:
> > * episode 1 generated with the trained RL method (trained to solve, not trained to explore),
> > * episode 2 with the meta-RL agent (learning from history + reflection of episode 1)
> >
> > If this method does just as well as running both episodes with the meta RL agent, then the claim that the meta RL agent is learning to explore is disproven: exploration wouldn't be more useful than purely learning to solve.
> >
> > If this method does worse, it really means that the meta-RL agent learned to explore.
> >
> > Would you agree with that?

---

> > > ### Author Response · Authors · 2025-11-29
> > >
> > > We thank the reviewer for the thoughtful follow-up and engaging discussion.
> > >
> > > ---
> > >
> > > **`"Learning to explore" vs "learning to try the task" in the first episodes.`**
> > >
> > > We thank the reviewer for the important question. We’d like to clarify that **"learning to explore" naturally induces "learning to try the task"**, the two components are not mutually exclusive.
> > >
> > > Suppose the agent generates two sequential episodes, $\tau_1$ and $\tau_2$, the first episode could lead to:
> > > 1.  $\tau_1$ successfully solves the task, or
> > > 2. $\tau_1$ does not solve the task but collects information that improves the probability of success in $\tau_2$
> > >
> > > If the objective of exploration is to improve the performance on $\tau_2$, then case (1) is strictly better than (2), and will be reinforced during training. Consequently,  **Attempting to solve the task in $\tau_1$ naturally emerges as part of the exploratory behavior** of the agent. This is true even when $\tau_1$ is not directly rewarded for its success (as the reviewer’s suggestion), because it will maximize the expected success rate on $\tau_2$.
> > >
> > > ### Discussion on the objective for exploration
> > >
> > > What incentivizes exploration is the **cross-episode credits assignment** presented in the paper, i.e., assigning a return $w_1 r_1 + w_2 r_2$ to $\tau_1$. As long as $w_2$ is >0, the agent is incentivized to gather more useful information at the first episode, even if $\tau_1$ is not successful.
> > >
> > > Therefore, there are two valid situations ($w_1 = 0, w_2 > 0$ and $w_1 > 0, w_2 > 0$) that are
> > > valid for the objective of “learning to explore”. We discuss both of them below:
> > >
> > > ### Situation 1: $w_1 = 0, w_2 > 0$ (suggested by the reviewer)
> > >
> > > The reviewer’s suggestion corresponds to the case where $w_1= 0$ and $w_2>0$ of the general formulation above:
> > > > If the agent is not rewarded to solve the game in early episodes more than in later ones? In that way, there would be no drive to solve episode 1, and this incentivize the agent to purely explore at the beginning?
> > >
> > > This design aligns with several seminal works of Meta-RL [1, 2, 3]. We agree that it would indeed induce “pure exploration” in the early episodes, but **this should not be the only valid design choice**.
> > >
> > > ### Situation 2:  $w_1 > 0, w_2 > 0$ (our choice in LaMer)
> > >
> > > In LaMer, we use the objective of cumulative rewards where $w_1>0, w_2>0$,
> > >  - **It also encourages exploration** since $w_2 > 0$. As discussed above, the key condition is $w_2 > 0$, not whether $w_1=0$. Meta-RL methods with cumulative rewards have also been studied in several previous works [4, 5, 6].
> > >  - **It provides a more flexible extension to the multi-episode settings**. Methods in situation 1 [1, 2, 3] need to explicitly distinguish which episodes are for exploration versus exploitation. In contrast, the objective of LaMer naturally generalizes to multi-episode setting, which is  more suitable for LLM-based agents.
> > >
> > > ---
> > >
> > > **`Experiment on “learning to explore” `**
> > >
> > > Following the reviewer’s suggestion, we conduct the experiments where the 1st episode is generated by the RL-trained agent, and the 2nd episode is generated by the Meta-RL trained agent. The results are shown below:
> > >
> > > *Table 1: Performance of the 2nd episode with varying sources of the 1st trajectory*
> > >  |        | Sokoban | MineSweeper | Webshop |
> > >  |--------|---------|-------------|---------|
> > >  | 1st RL + 2nd Meta-RL      | 45.3       | **70.3**  | 76.5       |
> > >  | 1st Meta-RL + 2nd Meta-RL | **46.9**   | 68.8      | **82.0**   |
> > >
> > > The result indicates the benefit of Meta-RL trained agents on Sokoban and Webshop. We also evaluate the performance with the 3rd episode where 1st and 2nd episodes are generated by the RL trained agent versus Meta-RL trained agent:
> > >
> > >  |        | Sokoban | MineSweeper | Webshop |
> > >  |---------|---------|-------------|---------|
> > >  | 1st/2nd RL + 3rd Meta-RL      | 46.9     |  74.2     | 75.9       |
> > >  | 1st/2nd Meta-RL + 3rd Meta-RL | **48.4** | **80.5** |**90.6**  |
> > >
> > > The 3d-episode result shows even stronger benefits of Meta-RL trained agents across all the environments. Overall, **these results provide a strong support of the explorative behavior of the trained agents** in LaMer.
> > >
> > >
> > > **Reference**
> > >
> > > [1] Some Considerations on Learning to Explore via Meta-Reinforcement Learning. NeurIPS, 2018.
> > >
> > > [2] Decoupling Exploration and Exploitation for Meta-Reinforcement Learning without Sacrifices. ICML, 2021.
> > >
> > > [3] First-Explore, then Exploit: Meta-Learning to Solve Hard Exploration-Exploitation Trade-Offs. NeurIPS, 2024.
> > >
> > > [4] RL^2: Fast Reinforcement Learning via Slow Reinforcement Learning. arXiv, 2016
> > >
> > > [5] VariBAD: A Very Good Method for Bayes-Adaptive Deep RL via Meta-Learning. ICLR, 2020.
> > >
> > > [6] Exploration in Approximate Hyper-State Space for Meta Reinforcement Learning. ICML, 2021.

---

### Official Review · Reviewer_73EN · 2025-10-28

**Soundness:** 4
**Presentation:** 3
**Contribution:** 3
**Rating:** 6
**Confidence:** 5

**Summary:**

This paper uses meta-reinforcement learning (Meta-RL) to train large language models (LLMs) as language agents across multiple game environments. The inner loop of Meta-RL is implemented as a simple reflection step, where the LLM is prompted to analyze and reflect on previous episodes. The outer loop is optimized using the GiGPO algorithm. The authors show that this approach improves performance compared to both RL and prompting-based baselines across various games. Their analysis further suggests that Meta-RL enhances the exploratory behavior of LLM agents, leading them to produce more diverse trajectories.

**Strengths:**

- The paper is well written and easy to follow, with clearly stated hypotheses and a clean experimental design. The idea is simple and promising to help improve LLM-agents to explore in RL tasks.

- It presents strong empirical results showing that meta-RL with reflection substantially improves the performance of LLM agents across multiple game environments.

**Weaknesses:**

- The paper does not analyze the reflections generated by the LLM. Do they make sense? Do they evolve over time? Is performance improvement mainly driven by changes in the policy or in the reflections themselves?
- The paper does not analyze the reflections generated by the LLM. Do they make sense? Do they evolve over time? Is performance improvement mainly driven by changes in the policy or in the reflections themselves?
- All experiments are conducted with a single LLM (Qwen3-4B). It would be important to test whether the results generalize across different model architectures and sizes.
- Lack of technical details:
    - The paper does not explain how trajectory diversity is computed.
    - Key training hyperparameters (e.g., batch size, learning rate, training budget) are not reported.
    - For procedurally generated environments such as Sokoban, it is unclear whether the same random seeds are used across episodes within a task—both in the RL and meta-RL settings.

**Questions:**

- Would your method also apply to more classical reasoning tasks where LLMs are trained with RL, such as MATH, BigBench, or similar benchmarks?

- Do you think your approach could work even without the reflection step — relying solely on in-context learning from previous trajectories provided in the prompt?

---

> ### Author Response · Authors · 2025-11-22
> **Rebuttal [1/2]**
>
> We thank the reviewer for the insightful comment and positive evaluation of our work. We are encouraged that the reviewer finds our paper well written and easy to follow,our idea simple but effective and our empirical results strong. We address the reviewer’s questions and concerns below.
>
> ---
>
> **`W1: Analyze the reflections generated by the LLM. Is performance improvement mainly driven by changes in the policy or in the reflections?`**
>
> We thank the reviewer for the valuable question. In our analysis, we observe two patterns in the model’s reflection:
> 1. **Informative reflections**, which provide a clear diagnosis of past mistakes and concrete, actionable plans for the next attempt. Below is an example for Sokoban:
>     > The player is not in a good position to get the box at X onto the target at Y without being blocked by the player's current position.  An improved plan is to let the player move to Z...
> 3. **Over-general reflections**, which tend to be vague and offer little guidance for adaptation, for example
>     > *To ensure all boxes are properly placed onto targets, we need to systematically address each one. The initial plan didn't account for … *
>
> We observed that both types of reflections appear in the base model. However, **LaMer explicitly trains the reflection step** using the reward from subsequent episodes, resulting in more *effective and helpful* reflections produced by the trained agents. Consequently, **the ability of self-reflection co-evolves with the policy**, jointly contributing to the performance improvement.
>
> ---
>
> **`W2: All experiments are conducted with a single LLM.`**
>
> Our method is designed as a general framework built on meta-RL principles and is model-agnostic. To validate this, we additionally ran experiments on Llama3.1-8B-Instruct, showing that our method works well on models in a different model architecture and model size. The new results are summarized as follows:
>
> *Table 1: Performance of proposed method and baseline on Llama-3.1-8B-Instruct*
> |       | Sokoban | MineSweeper |
> |-------|---------|-------------|
> | GiGPO | 6.3     | 34.4        |
> | LaMer | 20.3    | 65.6        |
>
>
> The results indicate that on Llama-3.1-8B-Instruct, LaMer still outperforms the RL baselines across the environments, demonstrating the general applicability of our method to different model architecture and size. We will include the results with the Llama-3.1-8B-Instruct in the revised manuscript.
>
> ---
>
> **`W3: Lack of technical details.`**
>
> Thanks for pointing that out! We kindly clarify the following:
> - **Trajectory diversity:** given each task instance, we sample multiple (e.g. $N=128$) trajectories from the model and construct the empirical discrete distribution over distinct trajectories. We then compute the entropy of this distribution to quantify the trajectory diversity. In Response to the reviewer’s feedback, we've also updated Figure 1 (right) with complementary visual example on a Minesweeper instance with the empirical probabilities of trajectories, to better illusrate this process.
>
> - **Training hyperparameters:** Below, we summarize the key training hyperparameters.
>
>     *Table 2: Training Hyperparameters*
>     |             | lr   | batch_size | epoch | max_resp_len | rollout_temp | val_temp | kl_coef | entr_coef |
>     |-------------|------|------------|-------|--------------|--------------|----------|---------|-----------|
>     | Sokoban     | 1e-6 | 16         | 300   | 1024         | 1.0          | 0.7      | 0.001   | 0.001     |
>     | MineSweeper | 1e-6 | 16         | 300   | 1024         | 1.0          | 0.7      | 0.001   | 0.001     |
>     | Webshop     | 1e-6 | 8          | 150   | 1024         | 1.0          | 0.7      | 0.001   | 0.001     |
>     | ALFWorld    | 1e-6 | 8          | 150   | 1024         | 1.0          | 0.7      | 0.001   | 0.001     |
> - **Random seeds for procedurally generated environments:** Yes, we use the same random seed (which leads to the same task instances in each training step) in both RL and meta-RL settings to ensure a fair comparison.
>
> In response to the reviewer's question, we have updated the manuscript with the training details in Appendix C.

---

> > ### Author Response · Authors · 2025-11-22
> > **Rebuttal [2/2]**
> >
> > **`Q1: Applicable to classical reasoning tasks such as MATH, BigBench?`**
> >
> > Yes, the LaMer framework could be applied to classical reasoning tasks with minor conceptual reinterpretation.
> > - For single-turn tasks like math problems, the "episode" would essentially be the entire solution generation and contain only a single turn.
> > - When process reward models are available, we could turn each solution generation into multiple intermediate steps with feedback received from the process reward model, which is analogous to the multi-turn setting we focus on.
> >
> > That being said, classical reasoning benchmarks have different challenges (primarily improving reasoning chains). Our work is complementary to these efforts, targeting distinct challenges (e.g. balancing exploration and exploitation, learning from experiences) in **interactive decision-making** tasks where:
> > - The agent receives environment feedback after each action.
> > - Episodes consist of multiple sequential interactions (observation → action → feedback loops).
> > - Strategic exploration is needed for the agent to gather environmental information.
> >
> > Investigating how much single-turn reasoning tasks benefit from our approach is an interesting direction for future research. In response to the reviewer’s feedback, we will discuss this in the revised manuscript.
> >
> > ---
> >
> > **`Q2: Does the approach work without the reflection step — relying solely on in-context learning from previous trajectories provided in the prompt?`**
> >
> > Thank you for the interesting question! We conduct the experiments comparing LaMer with 1) self-reflection and 2) only the previous trajectories. The results are shown below:
> >
> > *Table 3: Performance of LaMer with self reflection or only previous trajectories*
> > |                     | Sokoban | MineSweeper | Webshop |
> > |---------------------|---------|-------------|---------|
> > | LaMer (prev_trajs)  |  34.8   |   69.5      |  89.3   |
> > | LaMer (reflection)  |  55.9   |   74.4      |  89.1   |
> >
> >
> > The results show the benefit of self-reflection. LaMer with self-reflection consistently yields better performance compared to solely relying on the previous trajectories, particularly on Sokoban and MineSweeper. We’ll include the discussion in the revised manuscript.

---

### Official Review · Reviewer_3QcN · 2025-11-02

**Soundness:** 3
**Presentation:** 3
**Contribution:** 2
**Rating:** 6
**Confidence:** 2

**Summary:**

This paper introduces the LAMER meta-RL algorithm which makes agents explore better. The algorithm involves training the agent on sequences of related expisodes. After each algorithm there is a reflection step where the agent can write notes which are given to figure episodes. Since the meta-rl algorithm rewards the agent for reward across the entire sequence, the agent is incentivized to explore in earlier episodes and exploit in later ones.

They evaluate on Sokovan, Minesweeper, Alfworld, and Webshop, and find that LaMer produces both better exploration/diversity and higher overall reward.

**Strengths:**

The paper is well-motivated and is written clearly to explain how the authors are tackling the problem of exploration during RL.

While the Meta-RL framework itself is not novel, the application to LLMs is. The paper shows significant gains over single episode training across multiple benchmarks. The paper also shows out of distribution generalization compared to non meta learning on unseen benchmarks.

**Weaknesses:**

See the "questions" section.

Beyond this I would be interested in comparing against pass@k metrics for meta-exploration that have previously been explored in RL (for example, see Walder et. al Pass@K Policy Optimization: Solving Harder Reinforcement Learning Problems).  (I do not think this is necessary for this to be a good paper! Just a suggestion for extension.)

**Questions:**

- Why are there no seeds or error bars? RL experiments can have quite a lot of variance and it is typical to do at least 3 experimental seeds for each run.
- How much of the gain is attributed to the self reflection prompting versus the multi episode credit assignment?
   - Would be interesting to see an ablation of whether the reflection step is important or if merely showing the agent past episodes is sufficient.
- Qualitatively, did you notice the sampled trajectories to be more diverse? Entropy or similarity is an easily hackable objective. I'd also appreciate a bit more description on how exactly diversity was calculated, this was not entirely clear to me.

---

> ### Author Response · Authors · 2025-11-22
> **Rebuttal [1/2]**
>
> We thank the reviewer for a valuable and a positive evaluation of our work. We are glad that the reviewer found our work well-motivated, and our framework novel and effective. We address the reviewer’s questions below.
>
> ---
> **`Q1: Seeds and error bars.`**
>
> Due to the constraint of compute resources, in our initial submission, we did not estimate the variance of training across different random seeds. In response to the reviewer’s suggestion, we further conduct experiments with additional random seeds for the proposed method and the baseline. The results are shown below.
>
> *Table 1: Performance (mean ± std) of proposed method and baseline across 3 random seeds*
> |       | Sokoban      | MineSweeper  | Webshop      |
> |-------|--------------|--------------|--------------|
> | GiGPO | 39.6±3.4     | 52.3±2.5     | 72.8±1.8     |
> | LaMer | **49.6±4.4** | **76.6±1.8** | **91.8±2.0** |
>
> The results confirm that the training indeed exhibits variance across different random seeds, as noted by the reviewer. However, LAMER consistently outperforms the RL baseline across different seeds. **We will expand our experiment to include 3 random seeds for each method and report mean and variance** in the revised manuscript.
>
> ---
>
> **`Q2: Attribution of the gains to the self-reflection prompting vs the multi-episode credit assignment + ablation comparing using self-reflection and past-episode only.`**
>
> We thank the reviewer for raising an important point.
> - **Attribution of the gain**: Our insight is that multi-episode training incentivizes the agent to explore in early episodes, while self-reflection provides an efficient way of adaptation. In LaMer, **the two abilities are learned jointly, and both contribute to the overall performance gain**.
> - **Ablation on the reflection step**: Following the reviewer’s suggestion, we conduct an ablation of training the agents without self-reflection (only showing the past episodes to the agent). The results are shown below.
>
>     *Table 2: Performance of LaMer with and without self-reflection*
>     |                         | Sokoban | MineSweeper | Webshop |
>     |-------------------------|---------|-------------|---------|
>     | LaMer (Self-reflection) |  55.9   |   74.4      |  89.1   |
>     | LaMer (Past episodes)   |  34.8   |   69.5      |  89.3   |
>
>     The results confirm that self-reflection is an important contributor to the performance gain. LaMer with self-reflection yields better performance compared to only showing the agent with past episodes, particularly on Sokoban and MineSweeper.
>
> We’ll include the discussion in the revised manuscript.
>
> ---
>
> **`Q3: How exactly is diversity calculated? Are the sampled trajectories more diverse?`**
>
> **Diversity calculation**: The diversity is calculated as follows: for a given task instance, we sample $N$ (e.g., 128) trajectories from the agent. We then group identical trajectories together that have the same states and actions. With this, we obtain the empirical frequency of each distinct trajectory, for which we compute the entropy to quantify trajectory diversity.
>
> **Our observation on the diversity of trajectories**: Quantitatively, LaMer indeed produces more diverse trajectories compared to RL (Figure 1 left and Figure 3). In Response to the Reviewer’s feedback, we now update Figure 1 (right). The new visualization shows a complementary visual example on a Minesweeper instance with the empirical probabilities of trajectories. Additionally,we show the map of action (click) with RL (top) and MetaRL (bottom). This further demonstrates that LaMer induces more diverse trajectories.
>
> In response to the reviewer’s question, we haved updated Figure 1 and inlcude more explanation on the trajectory diversity in the manuscript.

---

> ### Author Response · Authors · 2025-11-22
> **Rebuttal [2/2]**
>
> ---
>
> **`W1: Suggestion about comparison against Pass@K Policy Optimization (Walder et. al, 2025)`**
>
> Thanks for the suggestion about comparison to the PKPO (Walder et al, 2025). It is an interesting work that maximizes pass@k objectives, which is also explored in [1]. The idea of maximizing reward across $k$ independent samples (instead of prioritizing individual sample performance) also induces diversity and exploration in the samples, which is relevant to our work.
>
> The key difference lies in whether the exploration can be leveraged in-context, making the two works suitable for different scenarios :
> - **PKPO**: Each trajectory is sampled *independently* and samples cannot leverage the environment feedback obtained by other samples. At test time, there is no sense of adaptation from past experiences. Therefore, it addresses how to train models to produce independent samples where at least one is good, and is suitable for tasks such as finding a good solution in a search space.
> - **LaMer**: Our method *jointly* trains explorative behavior and the adaptation capability given environment feedback. Trajectories are sampled *sequentially* with in-context adaptation at test. Therefore, it addresses how to train agents to actively explore environments and adapt from past experiences, and is suitable for interactive and sequential decision-making tasks where information needs to be gathered with early actions.
>
> This suggests that **the strengths of the two approaches are highly complementary** rather than competing; and combining the two can be highly beneficial. For example, applying PKPO on top of LaMer to optimize the pass@k of trials (where each trial contains multiple episodes as in our paper) is likely to incentivize diverse in-context adaptation strategies. We consider this as a promising direction for future work. In response to the Reviewer’s feedback, we will include a discussion of Walder et al.
>
> [1] Optimizing Language Models for Inference Time Objectives using Reinforcement Learning. Tang et al. ICML 2025.

---

### Author Response · Authors · 2025-12-03

Dear AC,

We would like to briefly summarize the rebuttal process for your convenience.

The reviewers’ feedback has been constructive and encouraging. Reviewers highlighted that our work is “*novel*” (3QcN, Qdy5), the proposed idea is ”*clear, simple, and promising*” (73EN, xsGO), and the empirical results are “*strong*” (3QcN, 73EN, Qdy5). Among the four reviewers:

Two reviewers engaged in the discussion:
 - **Reviewer Qdy5 (Rating 6 / Confidence 3)** acknowledged our response and kept the positive score
 - **Reviewer xsGo (Rating 6 / Confidence 4)** participated in the constructive discussion. We provided additional visualizations and experiment results supporting our claim of "learning to explore". The reviewer agreed that these results were "*valuable additions to the paper*".

Two reviewers did not have a chance to respond:
 - **Reviewer 3QcN (Rating 6 / Confidence 2)**
 - **Reviewer 73EN (Rating 6 / Confidence 5)**

For these reviewers, we've provided point-to-point clarifications and new results that strengthen the claims of the paper, including:
 - Clarification of the experimental details, connection to the related work and potential further application of our method.
 - Additional results on the ablation of self-reflection prompts and model choices (e.g. Llama-3.1-8B-Instruct).

We've updated the manuscript to incorporate the new results, visualizations and discussions. Thank you for your time! We're happy to provide any further clarifications if needed.

---

### Meta-Review · Area_Chair_7GG2 · 2026-01-06

**Summary:**

This work studies a new fine-tuning method for LLM-based agents that applies a meta-reinforcement learning framework, leveraging existing trajectory information through a self-reflection step rather than discarding past information as in standard reinforcement learning. Reviewers raised concerns regarding the lack of ablation studies on the self-reflection component, limited novelty, and computational efficiency. I would recommend that the authors provide additional empirical and analytical evidence to better address these issues.

**Reviewer Concerns:**

**Unaddressed issues:**

**Reviewer 3QcN:** none.

**Reviewer 73EN:** analysis of reflection quality.

**Reviewer xsGo:** none.

**Reviewer Qdy5:** limited novelty, sequential dependency, and computational inefficiency.

**Reviewer Scores:**

**Reviewer 3QcN:** 6 → 6

**Reviewer 73EN:** 6 → 6

**Reviewer xsGo:** 6 → 6

**Reviewer Qdy5:** 6 → 6

---

### Decision · Program_Chairs · 2026-01-26

Accept (Poster)